# Generalization Bounds for (Wasserstein) Robust Optimization

**Yang An**
Department of Mathematics
Columbia University
yangan@math.columbia.edu

**Rui Gao**
Department of IROM
University of Texas at Austin,
rui.gao@mccombs.utexas.edu

## Abstract

(Distributionally) robust optimization has gained momentum in machine learning community recently, due to its promising applications in developing generalizable learning paradigms. In this paper, we derive generalization bounds for robust optimization and Wasserstein robust optimization for Lipschitz and piecewise Hölder smooth loss functions under both stochastic and adversarial setting, assuming that the underlying data distribution satisfies transportation-information inequalities. The proofs are built on new generalization bounds for variation regularization (such as Lipschitz or gradient regularization) and its connection with robustness.

## 1 Introduction

Consider the following stochastic optimization problem of seeking a function $f : \mathscr{X} \to \mathbb{R}$ from the hypothesis space $\mathscr{F}$ so as to minimize the expected loss:

$$\min_{f \in \mathscr{F}} \left\{ \mathcal{L}^{\mathsf{true}}(f) := \mathbb{E}_{X \sim \mathbb{P}_{\mathsf{true}}}[f(X)] \right\}.$$

Here the data $X$ follows some underlying distribution $\mathbb{P}_{\mathsf{true}}$ on $\mathscr{X}$. In practice, $\mathbb{P}_{\mathsf{true}}$ is often replaced with its empirical estimate $\mathbb{P}_n = \frac{1}{n} \sum_{i=1}^{n} \boldsymbol{\delta}_{\hat{x}^i}$, where $\boldsymbol{\delta}$ is the Dirac measure:

$$\min_{f \in \mathscr{F}} \left\{ \mathcal{L}_n^{\mathsf{erm}}(f) := \mathbb{E}_{X \sim \mathbb{P}_n}[f(X)] \right\}.$$

The discrepancy between $\mathbb{P}_{\mathsf{true}}$ and $\mathbb{P}_n$ may be resulted from

(i) The sampling error of $\mathbb{P}_n$ when data are independently drew from $\mathbb{P}_{\mathsf{true}}$;

(ii) Non-i.i.d. data, e.g., $x^i$'s are samples from a Markov chain with a stationary distribution $\mathbb{P}_{\mathsf{true}}$;

(iii) The deploying environment is (adversarially) different from the data-collecting environment.

Such discrepancy largely contributes to the poor generalization behavior of empirical risk minimization. To tackle this issue, *robust optimization* [4, 37, 38] studies

$$\min_{f \in \mathscr{F}} \left\{ \mathcal{L}_{n,p}^{\mathsf{ro}}(f; \varrho) := \max_{\{x^i\}_{i=1}^n \in \mathscr{X}} \left\{ \frac{1}{n} \sum_{i=1}^{n} f(x^i) : \left( \frac{1}{n} \sum_{i=1}^{n} \|x^i - \hat{x}^i\|^p \right)^{\frac{1}{p}} \leq \varrho \right\} \right\}. \quad \text{(RO)}$$

Here $p \in [1, \infty]$; the uncertainty set, denoted by $\mathscr{X}_p(\varrho)$, describes the data perturbations to hedge against; and the radius $\varrho > 0$ specifies the level of robustness. In a similar spirit, *Wasserstein robust optimization* [15, 7, 19] studies

$$\min_{f \in \mathscr{F}} \left\{ \mathcal{L}_{n,p}^{\mathsf{wo}}(f; \varrho) := \max_{\mathbb{P}} \left\{ \mathbb{E}_{x \sim \mathbb{P}}[f(x)] : \mathcal{W}_p(\mathbb{P}, \mathbb{P}_n) \leq \varrho \right\} \right\}. \quad \text{(WO)}$$

Here the distributional uncertainty set, denoted by $\mathscr{P}_p(\varrho)$, consists of all probability distributions that are close to $\mathbb{P}_n$ in $p$-Wasserstein distance. When $p = \infty$, (WO) and (RO) coincide. Both problems have received increasing attentions in machine learning; see [25] for a survey.

35th Conference on Neural Information Processing Systems (NeurIPS 2021).

When data is generated stochastically, such as i.i.d. or Markovian (see (i)(ii) above), we would like to choose the radius $\varrho = \varrho_n$ adaptive to the sample size, so that the robust loss can balance between performance and robustness. More precisely, $\varrho_n$ should be large enough so that the robust loss dominates the true loss with high probability, but not be too large to produce an inferior solution. When the testing environment is adversarially different from the training environment (see (iii) above), consider the *adversarial loss* minimization

$$\min_{f \in \mathscr{F}} \left\{ \mathcal{L}_p^{\mathsf{adv}}(f; \varrho) := \max_{\mathbb{P} \in \mathscr{P}_p(\varrho)} \left\{ \mathbb{E}_{x \sim \mathbb{P}}[f(x)] : \mathcal{W}_p(\mathbb{P}, \mathbb{P}_{\mathsf{true}}) \leq \varrho \right\} \right\}, \tag{Adv}$$

where the radius $\varrho$ captures the difference between the training and testing environments. When $\mathbb{P}_{\mathsf{true}}$ is not available, a surrogate problem (WO) based on $\mathbb{P}_n$ is often considered. In this case, $\varrho$ is often fixed and we hope that the robust solution yielding from (WO) generalizes well.

The goal of this paper is to provide a comprehensive analysis of the generalization capability of the above robust learning paradigms. Specifically, in the stochastic setting, under proper choice of the radius $\varrho_n$, we derive finite-sample guarantees for (RO) and (WO). In the adversarial setting, we derive generalization bounds for (Adv) with a small fixed radius $\varrho$.

### Related Literature

*Stochastic setting.* The most related work to ours is [17], but we differ in several important ways. First, [17] studies only (WO) while we also consider (RO). Second, [17] studies i.i.d. data while we also consider non-i.i.d. by exploiting a different notion of transport inequality. Third, even in the i.i.d. setting, [17] studies only smooth losses and Wasserstein order $p = 1, 2$. We consider non-smooth losses and $p \in [1, 2]$, which includes important cases like $|\theta^\top x - y|^p$. Moreover, for $p = 1$, we tighten the order of the remainder; for $p = 2$, we obtain a cleaner expression using a different set of assumptions. All these extensions require in-depth analysis on worst-case scenarios.

For (WO) with i.i.d. data, [15] originally proposed to choose the radius $\varrho_n$ such that $\mathbb{P}_{\mathsf{true}}$ is contained in the uncertainty set with high probability, but the resulting performance bound is overly conservative and suffers from the curse of dimensionality. This issue has been mitigated asymptotically for smooth loss functions [6, 8, 33], and non-asymptotically for linear models [32, 12] and for smooth loss functions [17]. As mentioned above, our results apply to non-i.i.d. data and non-smooth losses.

For (RO) in the stochastic setting, its generalization properties were first studied in the pioneer work [39], which established algorithmic generalization generalization bounds for various learning tasks with i.i.d. and Markovian data. Nonetheless, many bounds developed therein suffer from the curse of dimensionality. This framework is then extended to metric learning [3] and ensemble models [41]. Our results circumvent the curse of dimensionality of the bounds developed in [39].

*Adversarial setting.* For $p \in [1, \infty)$, generalization properties of (WO) with fixed radius was considered first in [27]. The main difference is that we consider different magnitudes of data perturbation. We mainly focus on adversarial robustness where the radius is often a tiny number, and our results in Section 5 suggest that the generalization gap is small when the radius is small. While their motivation was from domain adaptation, and their bound is mostly useful when the Wasserstein radius is not extremely small, which suggests that the generalization gap becomes smaller as the radius grows to infinity (see their Remark 4). The bounds developed in [34, 30] does not depend on the Wasserstein radius, while our result indicates that when the radius is small, the remainder actually demonstrates its linear scaling and therefore tighter.

For $p = \infty$, [24, 40, 1] focus on deriving generalization bounds for specific loss function classes such as linear hypotheses and neural networks. Our results apply to generic function classes, and most importantly, the bound suggests the the generalization gap is controlled by not only the complexity of the loss function class, but also the complexity of the gradient norm functions of the loss functions. The intuition is that the gradient norm controls the robustness of the model subject to adversarial perturbations. As far as we know, the bound in such form is new in the literature. There are also some earlier works focusing on investigating the fundamental limit of adversarial learning and its difference from empirical risk minimization, such as [31, 35, 10, 11]. They consider somewhat different settings than ours, which usually involves a stylized data generating model that facilitates theoretical findings.

*Regularization.* Part of our analysis is based on the connection between robustness and regularization. The connection between (RO)(WO) and norm regularization was established for various

machine learning tasks [38, 37, 5, 6, 12], and the connection between (WO) and Lipschitz/gradient regularization has been established for smooth loss functions [32, 34, 16, 13, 8, 2] and non-smooth loss functions [18]. Our results improve the sandwich bounds in [18] for $p = 1$, and generalizes the results therein to piecewise Hölder smooth loss functions for $p \in (1, \infty]$. Finally, there are also works connecting regularization and distributional robustness under other choices of distance [26, 21, 14, 23].

## 2 Preliminaries

In this section, we prepare several technical tools that are useful in deriving our main results.

**Notations** For $p \in [1, \infty]$, we denote by $q$ its Hölder conjugate number, i.e. $\frac{1}{p} + \frac{1}{q} = 1$. Let $\mathscr{P}(\mathscr{X})$ be the set of Borel probability distributions on $\mathscr{X} = (\mathbb{R}^d, \|\cdot\|)$, and let $\mathscr{P}_p(\mathscr{X}) = \{\mathbb{P} \in \mathscr{P}(\mathscr{X}) : \mathbb{E}_{\mathbb{P}}[\|x\|^p] < \infty\}$. For a given probability measure $\mathbb{Q}$, denote by $(L_{\mathbb{Q}}^p(\mathscr{X}), \|\cdot\|_{p,\mathbb{Q}})$ the set of $L^p$ functions on $\mathscr{X}$ with respect to $\mathbb{Q}$. Unless stated otherwise, we will not assume the samples are i.i.d. Instead, we use $S_n = \{\hat{x}^i\}_{i=1}^n$ to represent a sample of size $n$, and denote by $\mu_{S_n}$ the sampling distribution and use $\mathbb{E}_{S_n}$ to indicate the expectation with respect to the $\mu_{S_n}$, and $\mathcal{L}^{\text{true}}(f) = \mathbb{E}_{S_n}[\frac{1}{n} \sum_{i=1}^n f(\hat{x}^i)]$. For a fixed set of $n$ points, we denote $\mathbb{P}_n$ as the empirical uniform distribution on these $n$ points. For two real numbers, $a$ and $b$, we denote $a \wedge b = \min(a, b)$, and $a \vee b = \max(a, b)$.

**Transport inequalities** In the stochastic setting, we will work with distributions that satisfy the following condition. Let $p \in [1, \infty)$ and $\tau > 0$. A distribution $\mathbb{Q} \in \mathscr{P}_p(\mathscr{X})$ satisfies a *transport inequality*, denoted as $T_p(\tau)$, if for all $\mathbb{P} \in \mathscr{P}_p(\mathscr{X})$ it holds that

$$\mathcal{W}_p^2(\mathbb{P}, \mathbb{Q}) \leq \tau \int_{\mathscr{X}} \log(d\mathbb{P}/d\mathbb{Q}) d\mathbb{P},$$

where $d\mathbb{P}/d\mathbb{Q}$ denotes the Radon-Nikodym derivative. The integral on the right-hand side of the inequality above represents the relative entropy between $\mathbb{P}$ and $\mathbb{Q}$. The transportation-information inequality $T_p(\tau)$ covers a variety of useful classes of distributions. For example, if $\mathscr{X}$ is bounded by $B$, then any distribution on $\mathscr{X}$ satisfies $T_1(8B^2)$. Gaussian distributions with covariance $\Sigma$ satisfies $T_2(2\lambda_{\max}(\Sigma))$. Interested readers are referred to [22] for a recent survey.

**Variation Regularization** Following [18], we define the *local* and *global slope* of a continuous function $f : \mathscr{X} \to \mathbb{R}$ at $x \in \mathscr{X}$, denoted by $|\partial f|(x)$ and $G(f)(x)$ respectively, as

$$|\partial f|(x) = \limsup_{\|\tilde{x} - x\| \to 0} \frac{(f(\tilde{x}) - f(x))_+}{\|\tilde{x} - x\|}, \quad G(f)(x) = \sup_{\tilde{x} \in \mathscr{X}} \frac{(f(\tilde{x}) - f(x))_+}{\|\tilde{x} - x\|}.$$

When $f$ is differentiable at $x$, we have $|\partial f|(x) = \|\nabla f(x)\|_*$. When $f$ is L-Lipschitz, we have $G(f)(x) \leq L$ for all $x \in \mathscr{X}$ and $\|G(f)\|_{\infty, \mathbb{P}_{\text{true}}} = \|f\|_{\text{Lip}}$. For a function $f$, denote by $\|f\|_{\text{Lip}}$ its Lipschitz norm whenever it exists, and by $\||\partial f|\|_q$ the $L^q(\mathbb{P}_{\text{true}})$-norm of its slope function. We define *variation regularization* as

$$\min_{f \in \mathscr{F}} \mathcal{L}_{n,q}^{\text{vr}}(f; \varrho) := \begin{cases} \frac{1}{n} \sum_{i=1}^n f(\hat{x}^i) + \varrho \||\partial f|\|_{q, \mathbb{P}_n}, & q < \infty, \\ \frac{1}{n} \sum_{i=1}^n f(\hat{x}^i) + \varrho \|f\|_{\text{Lip}}, & q = \infty. \end{cases} \tag{VR}$$

We introduce the following class of functions consisting of slope functions of loss functions.

$$\partial \mathscr{F}_q := \{|\partial f|^q : f \in \mathscr{F}\}, \quad \mathscr{N}_q := \left\{\frac{|\partial f|^q}{\||\partial f|\|_q^q} : f \in \mathscr{F}\right\}. \tag{1}$$

Throughout we utilize the following assumptions, which covers most loss functions that are amenable to optimize efficiently using first-order methods.

**Assumption 1.** *Every $f \in \mathscr{F}$ is L-Lipschitz.*

**Assumption 2.** *Every $f \in \mathscr{F}$ is piecewise differentiable. There exists $h, \alpha > 0$ such that for every $f \in \mathscr{F}$ and every $\tilde{x}, x$ in the same piece of $f$, it holds that $\|\nabla f(\tilde{x}) - \nabla f(x)\|_* \leq h\|\tilde{x} - x\|^\alpha$.*

# 3 Robustness and Variation Regularization

In this section, we establishes connection between robust optimization (RO) (WO) and variation regularization (VR), which serve as building blocks for results in Section 5.

We make the following assumption, which states that the loss function family has sufficient variation.

**Assumption 3.** $\eta := \inf_{f \in \mathscr{F}} \||\partial f|\|_{q, \mathbb{P}_{\text{true}}} > 0.$

We also define $\tilde{\eta} := \inf_{f \in \mathscr{F}} \||\partial f|\|_{q, \mathbb{P}_n}$. To simplify notations, we introduce a function $e_n : \mathbb{R}_+ \to \mathbb{R}_+$, defined as

$$e_n(\rho) = \sup_{f \in \mathscr{F}} \left\{ \mathbb{E}_{X \sim \mathbb{P}_n} \left[ (\rho - d(X, \mathscr{D}_f))_+ \right] \right\}, \tag{2}$$

which describes the data concentration within a margin of non-differentiable points.

**Remark 1.** $\tilde{\eta}$ can be ensured to be positive with high probability using Assumptions 1, 3 (see Lemma 5 in the supplementary). $e_n(\varrho)$ vanishes for differentiable families, and is often $\tilde{O}_p(1/n)$ when $\varrho = \varrho_n = \tilde{O}(1/\sqrt{n})$ (c.f., [18]).

Our first result in this section bounds the difference between (WO) and (VR) for $p > 1$.

**Proposition 1.** Let $p \in (1, \infty]$. Assume Assumptions 1, 2 and 3 hold. Then it holds that

$$|\mathcal{L}_{n,p}^{\text{wo}}(f; \varrho) - \mathcal{L}_{n,q}^{\text{vr}}(f; \varrho)| \leq \tilde{C}_0 \varrho^{(\alpha+1) \wedge p} + 2L e_n\big((pL/\tilde{\eta})^{\frac{1}{p-1}} \varrho\big), \quad \forall f \in \mathscr{F},$$

where $\tilde{C}_0$ equals $h$ if $p = \infty$, $h \max(1, (Lp/\tilde{\eta})^{\frac{\alpha+1}{p-1}})$ if $p \in (\alpha + 1, \infty)$, and $L(h/L)^{\frac{p-1}{\alpha}}$ otherwise.

Proposition 1 shows that (WO) and (VR) are close with a gap controlled by two high-order terms: one term $\varrho^{(\alpha+1) \wedge p}$ depends on the Hölder smoothness parameter $\alpha$; while the other term $e_n$ depends on the data distribution around the non-differentiable region. The smoother the loss function, the smaller the gap is. Intuitively, the worst-case distribution in (WO) perturbs the data approximately aligned with gradient ascent direction, thus the gradient norm in (VR) provides a tighter first-order approximation on the difference between the worst-case and the empirical losses for smoother losses. Compared to [18, Theorem 1], here we focus on Lipschitz loss functions, thereby we are able to obtain a cleaner result, but we have to modify the proof therein by bounding the worst-case loss differently.

Our second result in this section establishes a relationship between (WO) and (RO) for $p \in (1, \infty)$, recalling that they coincide for $p = \infty$.

**Proposition 2.** Let $p \in (1, \infty)$. Assume Assumptions 1, 2 and 3 hold. Then it holds that

$$0 \leq \mathcal{L}_{n,p}^{\text{wo}}(f; \varrho) - \mathcal{L}_{n,p}^{\text{ro}}(f; \varrho) \leq \frac{2L}{n} \big(\frac{L}{\lambda_n}\big)^{\frac{1}{p-1}}, \quad \forall f \in \mathscr{F},$$

where $\lambda_n$ is the dual minimizer of (WO) satisfying

$$(2^p - 1)\varrho^p \lambda_n \geq \tilde{\eta} \cdot \varrho - 2^{\alpha+1} \varrho^{\alpha+1} h - 2L e_n(2\varrho) - 2L e_n\big((Lp/\tilde{\eta})^{\frac{1}{p-1}} \varrho\big) - L\big(\frac{h}{L}\big)^{\frac{p-1}{\alpha}} \varrho^p - h(Lp/\tilde{\eta})\varrho^{\alpha+1}.$$

Proposition 2 shows that (WO) and (RO) have a uniformly upper bounded gap. Intuitively, both these two robust paradigms perturbs data points aligned with gradient ascent direction. The difference is that (WO) allows to perturb a tiny fraction of probability arbitrarily far away, but (RO) prohibits probability splitting. The proof is based on careful analysis on the probabilistic nature of the worst-case distribution. The first result of this kind was established in [19] for a single loss function, but the bound therein is not controlled uniformly for a family of losses. Under the conditions in Remark 1, the gap vanishes in the order $\tilde{O}_p(\varrho/n)$.

Thus far, we have shown that when $p \in (1, \infty]$, $\mathcal{L}_{n,q}^{\text{vr}}(f; \varrho_n)$, $\mathcal{L}_{n,p}^{\text{ro}}(\varrho_n)$ and $\mathcal{L}_{n,p}^{\text{wo}}(\varrho_n)$ only differ by a higher order gap that depends on the smoothness of the loss function class.

Next, we consider the case $p = 1$. In light of the example in [18], it is not always possible to have $\mathcal{L}_{n,\infty}^{\text{vr}}(\varrho) - \mathcal{L}_{n,1}^{\text{wo}}(\varrho)$ being of a higher order than $\varrho$. Below we identify conditions to achieve this. For $f \in \mathscr{F}$, we define cumulative distribution functions $H_f, \tilde{H}_f$ as

$$H_f(a) := \mathbb{P}_{\text{true}}\big\{\|f\|_{\text{Lip}} - G(f)(X) \leq a\big\}, \quad \tilde{H}_f(a) := \mathbb{P}_{\text{true}}\big\{\|f\|_{\text{Lip}} - |\partial f|(X) \leq a\big\}, \quad a \geq 0,$$

where the random variable $X$ has distribution $\mathbb{P}_{\text{true}}$. They describe how the global and local slope functions distribute around their maximum, i.e., the Lipschitz norm.

**Assumption 4.** *Assume there exists $\beta, \bar{a}, c > 0$ such that for all $a \in [0, \bar{a})$ and all $f \in \mathscr{F}$, it holds that $H_f(a^\beta) \geq ca$ for* (WO)*, and $\tilde{H}_f(a^\beta) \geq ca$ for* (RO)*.*

This assumption ensures sufficient concentration on the maximal slope $\|f\|_{\mathrm{Lip}}$, and large $\beta$ indicates that $G(f)(X)$ or $|\partial f|(X)$ concentrates more on the maximal slope. Ideally, we would like to have a large $\beta$, and in particular, $H_f$ has a density at 0 when $\beta = 1$. For example, if $H_f$ has a density near zero, then Assumption 4 imposes a positive lower bound on the density with $\beta = 1$. As another example, if $f$ is a linear function, then the set of global slopes is a singleton and $H_f(a) = \tilde{H}_f(a) = 1$ for all $a \geq 0$, hence Assumption 4 is trivially satisfied with $\bar{a} = \beta = \infty$. More generally, under the assumption that $\limsup_{\|x-\hat{x}\| \to \infty} \frac{(f(x)-f(\hat{x}))_+}{\|x-\hat{x}\|} = \|f\|_{\mathrm{Lip}}$, which is imposed by [17], the global slope $G(f)(x) = \|f\|_{\mathrm{Lip}}$ for all $x \in \mathscr{X}$, thereby $H_f(a) = 1$ for all $a \geq 0$.

**Proposition 3.** *Let $p = 1$. Assume Assumptions 1, 2, 4 hold. Let $\delta \in [0, \frac{1}{2})$. Then for every $f \in \mathscr{F}$, it holds that*
$$\mathcal{L}_{n,1}^{\mathrm{ro}}(f; \varrho) \leq \mathcal{L}_{n,1}^{\mathrm{wo}}(f; \varrho) \leq \mathcal{L}_{n,\infty}^{\mathrm{vr}}(f; \varrho),$$
*and that*
$$\mathcal{L}_{n,1}^{\mathrm{wo}}(f; \varrho) \geq \mathcal{L}_{n,\infty}^{\mathrm{vr}}(f; \varrho) - \epsilon_n, \quad \mathcal{L}_{n,1}^{\mathrm{ro}}(f; \varrho) \geq \mathcal{L}_{n,\infty}^{\mathrm{vr}}(f; \varrho) - \tilde{\epsilon}_n,$$
*where*
$$\epsilon_n = \varrho\Delta_n + 4hn^{\alpha\delta}\varrho^{\alpha+1} + 2Le_n(\varrho n^\delta),$$
$$\tilde{\epsilon}_n = \varrho\tilde{\Delta}_n + 4hn^{\alpha\delta}\varrho^{\alpha+1} + 2Le_n(\varrho n^\delta),$$
*where $\Delta_n$ (resp. $\tilde{\Delta}_n$) equals the $(\lfloor n^{1-\delta} \rfloor + 1)$-th order statistics of the sample $\{\|f\|_{\mathrm{Lip}} - G(f)(\hat{x}^i)\}_{i=1}^n$ (resp. $\{\|f\|_{\mathrm{Lip}} - |\partial f|(\hat{x}^i)\}_{i=1}^n$).*

**Remark 2.** *Consider data are generated i.i.d., $\varrho = \varrho_n = O(1/\sqrt{n})$, and $f$ has piecewise Lipschitz gradient (i.e., $\alpha = 1$). As shown in the supplementary, the remainder would be $O(n^{-\frac{\beta+1}{2}} + n^{-(\frac{1}{2}+\beta\delta)} + n^{-(1-2\delta)}) = O(n^{-\frac{\beta+1}{\beta+2}})$. When $G_f(X)$ has a positive mass on the maximal global slope, by setting $\delta = 0$, with high probability, we have $\Delta_n = 0$, and $\epsilon_n = O(\varrho^2 + e_n(\varrho))$, which is $\tilde{O}_p(1/n)$ under the condition in Remark 1.*

Proposition 3 shows that for $p = 1$, (RO), (WO) and (VR) are equivalent up to a higher order error term, which depends on the smoothness of the loss function as indicated by $\alpha$ and $e_n$, and the concentration of slopes as indicated by $\Delta_n$. The smoother the function and the more concentrated on the maximal global slope, the smaller the gap between (WO), (RO) and (VR) is. This improves the sandwich result of [18] in which $\epsilon_n = O(1/\sqrt{n})$. Our finer analysis is based on construction of feasible solutions for (WO) and (RO) obtained from perturbation of empirical data points as well as careful control on the distances of perturbation.

## 4 Generalization Bounds for Variation Regularization

In this section, we derive generalization bounds for variation regularization in the stochastic setting, which are building blocks to develop the generalization bounds for (WO) and (RO).

We make the following assumptions on the sampling distribution.

**Assumption 5.** *The sampling distribution $\mu_{S_n}$ satisfies $T_p(\tau_n)$.*

When $S_n$ contains i.i.d. samples from some underlying distribution $\mathbb{P}_{\mathrm{true}}$ that satisfies $T_p(\tau)$, then by tensorization of transport inequalities [22], $\mu_{S_n}$ satisfies $T_p(\tau_n)$ with $\tau_n = \tau n^{2/p-1}$. In Example 2, we will consider Markovian data that satisfies $T_1(\tau_n)$ with $\tau_n = O(n)$.

Suppose we work with a parametric family $\mathscr{F} = \{f_\theta\}_{\theta \in \Theta}$, where the parameter space $\Theta$ is endowed with a norm $\|\cdot\|_\Theta$. We impose the following assumption of a Lipschitz parametric family.

**Assumption 6.** *There exists $\kappa > 0$ such that $|f_{\theta'}(x) - f_\theta(x)| \leq \kappa\|\theta' - \theta\|$ for all $x \in \mathscr{X}$.*

**Proposition 4.** *Let $p = 1$. Assume Assumptions 1, 5 and 6 are in force. Let $t > 0$. Set*
$$\varrho_n = \sqrt{\frac{\tau_n(t + \log\mathcal{N}(\frac{1}{n}; \Theta, \|\cdot\|_\Theta))}{n^2}}.$$

*Then with probability at least $1 - e^{-t}$, for all $f \in \mathscr{F}$, it holds that*

$$\mathcal{L}^{\mathsf{true}}(f) \leq \mathcal{L}^{\mathsf{vr}}_{n,\infty}(f; \varrho_n) + \frac{2\kappa}{n}.$$

Proposition 4 shows that by choosing the radius properly, Lipschitz regularization upper bounds the true risk $\mathcal{L}^{\mathsf{true}}(f)$ uniformly with high probability up to some error terms. The radius $\varrho_n$ depends on the transport inequality parameter of sampling distribution in Assumption 5 and the complexity of the loss function class. When $\tau_n = O(n)$ and $\log \mathcal{N}(\frac{1}{n}; \Theta, \|\cdot\|_\Theta) = \tilde{O}(1)$, we have $\varrho_n = \tilde{O}(1/\sqrt{n})$, corresponding to the canonical choice.

For $p > 1$, let us assume $\mathbb{P}_{\mathsf{true}}$ is continuous and define the population counterpart of $e_n(\varrho)$ in (2) as

$$e(\rho) = \sup_{f \in \mathscr{F}} \left\{ \mathbb{E}_{X \sim \mathbb{P}_{\mathsf{true}}} \left[ (\rho - d(X, \mathscr{D}_f))_+ \right] \right\}. \tag{3}$$

**Proposition 5.** *Let $p \in (1, 2]$. Assume Assumptions 1, 2, 3, 5 and 6 are in force. Let $t > 0$. Set*

$$\varrho_n = \left( 1 - 2\mathfrak{R}_n(\mathscr{N}_q) - (L/\eta)^q \sqrt{\frac{t}{2n}} \right)^{-\frac{1}{q}} \sqrt{\frac{\tau_n(t + \log(1 + \mathcal{N}(\frac{1}{n}; \Theta, \|\cdot\|_\Theta)))}{n^{2/p}}}.$$

*Then with probability at least $1 - e^{-t}$, for all $f \in \mathscr{F}$, it holds that*

$$\mathcal{L}^{\mathsf{true}}(f) \leq \mathcal{L}^{\mathsf{vr}}_{n,q}(f; \varrho_n) + \frac{2\kappa}{n} + C_0 \varrho_n^{(\alpha+1)\wedge p} + 2Le\left( (pL/\eta)^{\frac{1}{p-1}} \varrho_n \right),$$

*where $C_0$ equals $h$ if $p = \infty$, $h \max(1, (Lp/\eta)^{\frac{\alpha+1}{p-1}})$ if $p \in (\alpha + 1, \infty)$, and $L(h/L)^{\frac{p-1}{\alpha}}$ otherwise.*

Recalling $\mathscr{N}_q$ is the set of normalized slope defined in (1), and the inflation of the radius compared with the case of $p = 1$ is to compensate the discrepancy between $\||\partial f|\|_q$ and $\||\partial f|\|_{q,\mathbb{P}_n}$. The inflation factor is well-defined so long as the Rademacher complexity of $\mathscr{N}_q$ diminishes with respect to sample size. The additional remainder term is similar to the one in Proposition 1, which measures the difference between (WO) and (VR) when the nominal distribution is $\mathbb{P}_{\mathsf{true}}$. The proof of Propositions 4 and 5 follows a similar flow as that of [17, Corollary 2], but our result extends it to non-i.i.d. and non-smooth settings.

**Example 1** (Linear Prediction with Polynomial Loss)**.** *Consider a supervised learning problem with a feature vector $x \in \mathscr{X} \subset \mathbb{R}^d$ and a response variable $y \in \mathscr{Y} \subset \mathbb{R}$. Suppose $\|x\|_2 \leq B_1$ for all $x \in \mathscr{X}$ and $|y| \leq B_2$ for all $y \in \mathscr{Y}$. Let $\Theta$ be a bounded set in $\mathbb{R}^d$ such that $\|\theta\|_2 \leq B_3 - 1$ for all $\theta \in \Theta$. Consider a linear predictor with polynomial loss $\mathscr{F} = \{(x, y) \mapsto f_\theta(x, y) = |\theta^\top x - y|^p : \theta \in \Theta\}$. We now discuss how to ensure the assumptions in Propositions 4, with details in the supplementary. It is easy to verify that every $f_\theta(x, y)$ is Lipschitz in both variables and parameters, with $\kappa = p(B_1 + B_2)^p B_3^{p-1}$ in Assumption 6. Suppose the samples are i.i.d. from $\mathbb{P}_{\mathsf{true}}$ satisfying $T_p(\tau)$, which implies Assumption 5 with $\tau_n = \tau n^{\frac{2}{p}-1}$. Hence we can apply Proposition 4, and by [36, Example 5.8], we have $\log \mathcal{N}(\frac{1}{n}; \Theta, \|\cdot\|_\Theta) \leq d \log(1 + 2B_3 n)$.*

**Example 2** (Markovian Data)**.** *In this example, we show that our Assumption 5 about $\mu_{S_n}$ can be satisfied for Markovian data. Consider a homogeneous Markov chain on $\mathscr{X}$ with a transition kernel $P(dy|x)$. Assume there exist $\alpha_0 > 0, \bar{\beta} < \alpha_0$ and $C < \infty$, such that*

$$\int_{\mathscr{X}} e^{\alpha_0 \|y\|^2} P(dy|x) \leq C e^{\bar{\beta}\|x\|^2}, \quad \forall x \in \mathscr{X}.$$

*Let $\{x^i\}_{i=1}^n$ be $n$ consecutive outputs of this Markov chain. Then by Theorem 4.1 in [9], $\mu_{S_n}$ satisfies $T_1(\tau_n)$ with $\tau_n = \frac{2n(1 + \log C)}{\alpha_0 - \bar{\beta}}$. Thus Assumption 5 is satisfied.*

## 5 Generalization Bounds for (Wasserstein) Robust Optimization

In this section, we develop the generalization bounds for (WO) and (RO), based on the results in previous sections.

## 5.1 Stochastic Setting

Combining the results in Sections 3 and 4, we immediately obtain the following theorems.

**Theorem 1.** *Let $p \in (1, 2]$ and $t > 0$. Under the setup of Propositions 1 and 5, with probability at least $1 - e^{-t}$, for all $f \in \mathscr{F}$, it holds that*

$$\mathcal{L}^{\mathsf{true}}(f) \leq \mathcal{L}_{n,p}^{\mathsf{wo}}(f; \varrho_n) + \epsilon_n,$$

*where*

$$\epsilon_n = (C_0 + \tilde{C}_0)\varrho_n^{(\alpha+1)\wedge p} + 2Le_n\big((pL/\tilde{\eta})^{q-1}\varrho_n\big) + 2Le\big((pL/\eta)^{q-1}\varrho_n\big) + \frac{2\kappa}{n}.$$

*With the additional setup as in Proposition 2, it holds that*

$$\mathcal{L}^{\mathsf{true}}(f) \leq \mathcal{L}_{n,p}^{\mathsf{ro}}(f; \varrho_n) + \epsilon_n + \frac{2L}{n}\Big(\frac{L}{\lambda_n}\Big)^{\frac{1}{p-1}}.$$

**Theorem 2.** *Let $p = 1$ and $t > 0$. Under the setup of Propositions 3 and 4, with probability at least $1 - e^{-t}$, for all $f \in \mathscr{F}$, it holds that*

$$\mathcal{L}^{\mathsf{true}}(f) \leq \mathcal{L}_{n,1}^{\mathsf{wo}}(f; \varrho_n) + \epsilon_n + \frac{2\kappa}{n},$$

$$\mathcal{L}^{\mathsf{true}}(f) \leq \mathcal{L}_{n,1}^{\mathsf{ro}}(f; \varrho_n) + \tilde{\epsilon}_n + \frac{2\kappa}{n}.$$

Both Theorems 1 and 2 show that by choosing a reasonably small radius without suffering from the curse of dimensionality, the robust solution resulting from solving (VR), (RO), or (WO) has a nice generalization bound, expressed as the robust loss plus a high-order term. The high-order term depends mainly on the smoothness of the loss function as indicated by $\alpha$, the (non-)concentration of probability distributions on non-differentiable points as indicated by $e_n$, and the concentration of the maximal slope as indicated by $\Delta_n$ (resp. $\tilde{\Delta}_n$) in the expression of $\epsilon_n$ (resp. $\tilde{\epsilon}_n$) (when the Wasserstein order $p = 1$).

**Example 3** (Unsupervised Learning). *Consider the principal component analysis*

$$\max_{W \in \mathbb{R}^{d \times k}} \big\{ \|W^\top X\|_F^2 : \ W^\top W = I_k \big\},$$

*that seeks for $d$ principal directions, along which the data has maximized variance. Here $X$ is a $d \times n$ sample matrix consisting of $n$ samples $\{X_i\}_{i=1}^n$ from $\mathbb{P}_{\mathsf{true}}$ on a set $\mathscr{X} \subset \{x \in \mathbb{R}^d : \|x\|_2 \leq B\}$; $\|\cdot\|_F$ denotes the Frobenius norm; $I_k$ denotes the $k$-dimensional identity matrix; and $W = [w_1, \ldots w_k]$ is the set of $k$ orthonormal principal directions in $\mathbb{R}^d$. Assume that $\mathbb{P}_{\mathsf{true}}$ satisfies $T_2(\tau)$, $\mathbb{E}_{x \sim \mathbb{P}_{\mathsf{true}}}[x] = 0$, and the smallest eigenvalue of the covariance matrix $\mathbb{E}_{x \sim \mathbb{P}_{\mathsf{true}}}[xx^\top]$ is positive. Let $p = 2$ and $\mathscr{F} = \{x \mapsto f_W(x) = -\|W^\top x\|_2^2 : W^\top W = I_k\}$. Below we verify Assumptions 1, 2, 3, 5, 6 as required by Theorem 1 and compute the constants, and refer to the supplementary for detailed calculation.*

*It is easy to verify that Assumption 1 since $f_W$ is $2Bk$-Lipschitz; Assumption 2 holds with $\alpha = 1$; the zero-mean and non-singular covariance of $\mathbb{P}_{\mathsf{true}}$ imply Assumption 3; Assumption 5 holds with $\tau_n = \tau$; and Assumption 6 holds with $\kappa = 2\sqrt{k}B^2$. Finally, to compute the covering number of the set $\mathcal{W} = \{W : W^\top W = I_k\}$, observe that each $W$ consists of $k$ orthogonal vectors, each of which belongs to the unit ball in $\mathbb{R}^d$. Hence $\log \mathcal{N}(\frac{1}{n}; \mathcal{W}, \|\cdot\|_2) \leq kd \log((1+2n))$. Thus, we improve the bound in [39, Example 8] which has an exponential dependence on $d$.*

Before closing this subsection, we would like to emphasize that this stochastic setting is drastically different from the adversarial setting in the next subsection, for which many recent studies on the generalization properties of robust optimization. In fact, choosing a good radius scaling scheme with nice finite-sample guarantees is at the core of Wasserstein DRO and remains open for quite some time until the recent work [17]. We generalize this result to non-i.i.d. and non-smooth settings as well as for (RO).

## 5.2 Adversarial Setting

In the adversarial setting (Adv), we are trying to minimize the expected loss under the worst-case scenario when the learning algorithm is being purposely attacked by an adversary. By replacing $\mathbb{P}_{\text{true}}$ with $\mathbb{P}_n$ in (Adv), we obtain the empirical adversarial robustness problem, which is the same as the Wasserstein robust optimization problem (WO). One of the most fundamental questions in adversarial robustness problem is to estimate the generalization gap between (Adv) and its empirical counterpart (WO). We aim to derive generalization bounds for $p \in [1, \infty]$ by virtue of Theorems 1 and 3. Following the literature, in this subsection, we assume $\mathbb{P}_n$ consists of i.i.d. samples from a continuous distribution $\mathbb{P}_{\text{true}}$.

We start with $p = 1$.

**Theorem 3.** *Let $p = 1$. Under the setup of Proposition 3, assume that $\|f\|_\infty \leq M$ for all $f \in \mathscr{F}$. Let $\delta \in [0, \frac{1}{2})$ and $t > 0$. Then with probability at least $1 - e^{-t}$, it holds that for every $f \in \mathscr{F}$,*

$$|\mathcal{L}_{n,1}^{\text{wo}}(f; \varrho) - \mathcal{L}_1^{\text{adv}}(f; \varrho)| \leq 2\mathfrak{R}_n(\mathscr{F}) + M\sqrt{\frac{t}{2n}} + 2\epsilon_n + C\varrho^{1 + \frac{\alpha\beta}{\alpha+\beta}}.$$

On the right-hand side of the inequality in Theorem 3, the first two terms are identical to the generalization bound of empirical risk minimization. The third term $\epsilon_n$ from Theorem 3 is linear in the adversarial perturbation level $\varrho$, which represents the inflation of generalization bound in the adversarial setting as opposed to the stochastic setting, whose dominating factor is the concentration of the maximal slope. The rest terms are of higher-order for small adversarial perturbation level $\varrho$, that mainly depends on the smoothness of the loss function.

Next, we consider $p \in (1, \infty]$.

**Theorem 4.** *Let $p \in (1, \infty]$. Under the setup of Proposition 1, assume that $\|f\|_\infty \leq M$ for all $f \in \mathscr{F}$. Let $\varrho \leq c^{\frac{1}{\delta}} \left(\frac{\bar{a}}{2}\right)^{\frac{1}{\beta\delta}}$ and $t > 0$. Then with probability at least $1 - 2e^{-t}$, it holds that for every $f \in \mathscr{F}$,*

$$|\mathcal{L}_{n,p}^{\text{wo}}(f; \varrho) - \mathcal{L}_p^{\text{adv}}(f; \varrho)| \leq 2\mathfrak{R}_n(\mathscr{F}) + M\sqrt{\frac{t}{2n}} + \frac{2\mathfrak{R}_n(\partial\mathscr{F}_q) + L^q\sqrt{\frac{t}{2n}}}{q \cdot (\eta \wedge \tilde{\eta})^{\frac{1}{p-1}}} \cdot \varrho$$
$$+ C_0\varrho^{(\alpha+1)\wedge p} + Le_n\left((pL/\tilde{\eta})^{\frac{1}{p-1}}\varrho\right) + C\varrho^2.$$

Similar to the case of $p = 1$, the first two terms in the above bound is identical to the bound of empirical risk minimization, and the high-order term in the second row depends on the smoothness of the loss functions. The major difference from $p = 1$ is that, the inflation term in the adversarial setting depends on $\mathfrak{R}_n(\partial\mathscr{F}_q)$, the Rademacher complexity of the class of gradient norm functions. This term appears to be new in the literature. Intuitively, if the complexity of the variation is small, the model is more robust to the adversarial perturbations and thus generalizes better.

Theorems 3 and 4 are proved based on a different strategy than that in the stochastic setting. Set $\mathcal{L}_q^{\text{vr}}(\varrho) := \min_{f \in \mathscr{F}} \{\mathbb{E}_{\mathbb{P}_{\text{true}}}[f] + \varrho\|\|\partial f\|\|_{q, \mathbb{P}_{\text{true}}}\}$. Using the triangle inequality we decompose the difference $|\mathcal{L}_{n,p}^{\text{wo}}(f; \varrho) - \mathcal{L}_p^{\text{adv}}(f; \varrho)|$ into three terms $|\mathcal{L}_{n,p}^{\text{wo}}(f; \varrho) - \mathcal{L}_{n,q}^{\text{vr}}(f; \varrho)|$, $|\mathcal{L}_{n,q}^{\text{vr}}(f; \varrho) - \mathcal{L}_q^{\text{vr}}(f; \varrho)|$, $|\mathcal{L}_q^{\text{vr}}(f; \varrho) - \mathcal{L}_p^{\text{adv}}(f; \varrho)|$ and then bound each term individually.

## 6  Concluding Remarks

In this paper, we thoroughly analyzed the finite-sample performance bounds for robust optimization, drawing connection with transport inequality and local Rademacher complexity. Our results generalize existing results from various perspectives, and particularly, to non-smooth loss functions and to non-i.i.d. data, both of which requires non-trivial analysis. Our results help to better understand the regularization effect of robust learning and explain their superior empirical performances. We hope our results can inspire regularization schemes for other applications as well. Our bounds involve the Rademacher complexities of (normalized) gradient norm function $\partial\mathscr{F}_q, \mathscr{N}_q$ and margin to discontinuity $\mathscr{I}_\varrho$, which are non-standard; we have provided various examples and will investigate in more depth for future work.

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
