# Supplementary Material

## A  Auxiliary Results

**Lemma 1** ([19, 20]). *Suppose every $f \in \mathscr{F}$ is $L$-Lipschitz continuous. When $p \in [1, \infty)$, it holds that*

$$\mathcal{L}_{n,p}^{\mathsf{wo}}(f; \varrho) = \min_{\lambda \geq 0} \left\{ \lambda \varrho^p + \frac{1}{n} \sum_{i=1}^{n} \sup_{x \in \mathscr{X}} \left\{ f(x) - \lambda \|x - \hat{x}^i\|^p \right\} \right\}.$$

*When $p = \infty$, it holds that*

$$\mathcal{L}_{n,\infty}^{\mathsf{wo}}(f; \varrho) = \mathcal{L}_{n,\infty}^{\mathsf{ro}}(f; \varrho) = \frac{1}{n} \sum_{i=1}^{n} \sup_{x \in \mathscr{X}, \|x - \hat{x}^i\| \leq \varrho} f(x).$$

**Lemma 2** (Lemma 2 in [29]).

$$\mathbb{E}_S \left[ \sup_{f \in \mathscr{F}} \left( \mathbb{E}_{\mathbb{P}_{\mathsf{true}}}[f] - \mathbb{E}_n[f] \right) \right] \leq 2 \mathfrak{R}_n(\mathscr{F}).$$

**Lemma 3.** *Suppose $f(x) \in [0, M]$ for all $f \in \mathscr{F}$, then for any $t > 0$, with probability $1 - e^{-t}$, we have:*

$$\sup_{f \in \mathscr{F}} |\mathbb{E}_{\mathbb{P}_{\mathsf{true}}}[f] - \mathbb{E}_{\mathbb{P}_n}[f]| \leq 2 \mathfrak{R}_n(\mathscr{F}) + M \sqrt{\frac{t}{2n}}.$$

*If we have $\|f\|_\infty < M$, then by replacing $f$ by $f + M$, we can get the same result by replacing $M$ by $2M$.*

**Lemma 4** (Contraction Lemma). *Let $f$ be a $L$-Lipschitz function, and $\mathcal{F}$ a family of functions on $\mathscr{X}$. Then:*

$$\mathfrak{R}_n(f \circ \mathscr{F}) \leq L \cdot \mathfrak{R}_n(\mathscr{F}),$$

*where $f \circ \mathscr{F} = \{f \circ g : g \in \mathscr{F}\}$.*

We bound $\|\|\partial f\|\|_{q, \mathbb{P}_n}$ and $\|\|\partial f\|\|_{q, \mathbb{P}_{\mathsf{true}}}$ in both absolute difference and relative difference.

**Lemma 5.** *Assume Assumptions 1 and 3 hold. Then*

$$\left| \|\|\partial f\|\|_{q, \mathbb{P}_n} - \|\|\partial f\|\|_{q, \mathbb{P}_{\mathsf{true}}} \right| \leq \frac{1}{q} (\eta \wedge \tilde{\eta})^{-\frac{q}{p}} \left| \|\|\partial f\|\|_{q, \mathbb{P}_n}^q - \|\|\partial f\|\|_{q, \mathbb{P}_{\mathsf{true}}}^q \right|.$$

*Let $t > 0$. Then with probability at least $1 - e^{-t}$,*

$$\|\|\partial f\|\|_q \leq \|\|\partial f\|\|_{q, \mathbb{P}_n} \left( 1 - 2 \mathfrak{R}_n(\mathscr{N}_q) - (L/\eta)^q \sqrt{\frac{t}{2n}} \right)^{-\frac{1}{q}}, \quad \forall f \in \mathscr{F}.$$

*Proof.* For the first part, notice that the function $s \mapsto s^{\frac{1}{q}}$ is Lipschitz continuous on $((\eta \wedge \tilde{\eta})^q, \infty)$ with constant no larger than $\frac{1}{q}(\eta \wedge \tilde{\eta})^{-\frac{q}{p}}$. The result follows from the Mean Value Theorem.

For the second part, using McDiarmid's inequality, with probability at least $1 - e^{-t}$, for every $f \in \mathscr{F}$,

$$\sup_{f \in \mathscr{F}} \left| \mathbb{E}_{\mathbb{P}_n} \left[ \frac{|\partial f|(x)^q}{\|\|\partial f\|\|_{\mathbb{P}_{\mathsf{true}}, q}^q} \right] - 1 \right| \leq \mathbb{E}_{S_n} \left[ \sup_{f \in \mathscr{F}} \left| \mathbb{E}_{\mathbb{P}_n} \left[ \frac{|\partial f|(x)^q}{\|\|\partial f\|\|_{\mathbb{P}_{\mathsf{true}}, q}^q} \right] - 1 \right| \right] + (L/\eta)^q \sqrt{\frac{t}{2n}},$$

which implies that

$$\frac{\|\|\partial f|(x)\|_{\mathbb{P}_n, q}^q}{\|\|\partial f|(x)\|_{\mathbb{P}_{\mathsf{true}}, q}^q} - 1 \geq -2 \mathfrak{R}_n(\partial \mathscr{F}_q) - (L/\eta)^q \sqrt{\frac{t}{2n}}.$$

Thus, it holds that

$$\|\|\partial f\|\|_{\mathbb{P}_{\mathsf{true}}, q} \leq \|\|\partial f\|\|_{\mathbb{P}_n, q} \left( 1 - 2 \mathfrak{R}_n(\partial \mathscr{F}_q) - (L/\eta)^q \sqrt{\frac{t}{2n}} \right)^{-\frac{1}{q}}.$$

$\square$

# B  Proofs for Section 3

## B.1  Proof of Proposition 1

By [18, Lemma EC.8], for all $\delta \geq 0$,

$$\big| \sup_{x \in \mathscr{X}, \|x - \hat{x}^i\| \leq \delta} f(x) - f(\hat{x}^i) - |\partial f|(\hat{x}^i)\delta \big| \leq h\delta^{\alpha+1} + 2L\big(\delta - d(\hat{x}^i, \mathscr{D}_f)\big)_+. \tag{4}$$

We start with the simple case of $p = \infty$, for which (WO) and (RO) coincide. Using Lemma 1, we have

$$\mathcal{L}_{n,\infty}^{\mathsf{wo}}(f; \varrho) = \mathcal{L}_{n,\infty}^{\mathsf{ro}}(f; \varrho) = \frac{1}{n} \sum_{i=1}^{n} \sup_{x \in \mathscr{X}, \|x - \hat{x}^i\| \leq \varrho} f(x).$$

By (2) and (4),

$$\big| \mathcal{L}_{n,\infty}^{\mathsf{wo}}(f; \varrho) - \mathcal{L}_{n,q}^{\mathsf{vr}}(f; \varrho) \big| \leq h\varrho^{\alpha+1} + 2Le_n(\varrho).$$

Next we consider $p \in (1, \infty)$. Let us first prove for the upper bound. By Assumption 1,

$$0 \leq \sup_{x^i \in \mathscr{X}} \{ f(x^i) - f(\hat{x}^i) - \lambda \|x^i - \hat{x}^i\|^p \} \leq \sup_{\delta \geq 0} \{ L\delta - \lambda\delta^p \},$$

hence the optimal $\delta$ satisfies

$$\delta \leq \left( \frac{L}{\lambda} \right)^{\frac{1}{p-1}}. \tag{5}$$

If $p < 1 + \alpha$, by (4) we can bound the inner maximization as

$$\sup_{x^i} \{ f(x^i) - f(\hat{x}^i) - \lambda \|x^i - \hat{x}^i\|^p \}$$

$$= \sup_{0 \leq \delta \leq (\frac{L}{\lambda})^{\frac{1}{p-1}}, \|x^i - x\| \leq \delta} \{ f(x^i) - f(\hat{x}^i) - \lambda\delta^p \}$$

$$\leq \sup_{0 \leq \delta \leq (\frac{L}{\lambda})^{\frac{1}{p-1}}} \{ |\partial f|(\hat{x}^i)\delta + h\delta^{\alpha+1} + 2L\big(\delta - d(\hat{x}^i, \mathscr{D}_f)\big)_+ - \lambda\delta^p \}$$

$$\leq \sup_{0 \leq \delta \leq (\frac{L}{\lambda})^{\frac{1}{p-1}}} \{ |\partial f|(\hat{x}^i)\delta - (\lambda - h\delta^{\alpha+1-p})\delta^p + 2L\big(\delta - d(\hat{x}^i, \mathscr{D}_f)\big)_+ \}$$

$$\leq \sup_{\delta \geq 0} \{ |\partial f|(\hat{x}^i)\delta - (\lambda - h(\tfrac{L}{\lambda})^{\frac{\alpha+1-p}{p-1}})\delta^p \} + 2L\big( (\tfrac{L}{\lambda})^{\frac{1}{p-1}} - d(\hat{x}^i, \mathscr{D}_f) \big)_+,$$

which can be finite only when $\lambda \geq L(\frac{h}{L})^{\frac{p-1}{\alpha}}$. It follows from Lemma 1 that

$$\mathcal{L}_{n,p}^{\mathsf{wo}}(f; \varrho) - \frac{1}{n} \sum_{i=1}^{n} f(\hat{x}^i)$$

$$\leq \min_{\lambda \geq L(\frac{h}{L})^{\frac{p-1}{\alpha}}} \left\{ \lambda\varrho^p + \frac{1}{n} \sum_{i=1}^{n} \sup_{\delta_i > 0} \{ |\partial f|(\hat{x}^i)\delta_i - (\lambda - h(\tfrac{L}{\lambda})^{\frac{\alpha+1-p}{p-1}})\delta_i^p \} + 2L\big( (\tfrac{L}{\lambda - L(\frac{h}{L})^{\frac{p-1}{\alpha}}})^{\frac{1}{p-1}} - d(\hat{x}^i, \mathscr{D}_f) \big)_+ \right\}$$

$$\overset{\lambda \leftarrow \lambda - L(\frac{h}{L})^{\frac{p-1}{\alpha}}}{=} \min_{\lambda \geq 0} \left\{ \lambda\varrho^p + \frac{1}{n} \sum_{i=1}^{n} \sup_{\delta_i > 0} \{ |\partial f|(\hat{x}^i)\delta_i - \lambda\delta_i^p \} + 2L\big( (\tfrac{L}{\lambda})^{\frac{1}{p-1}} - d(\hat{x}^i, \mathscr{D}_f) \big)_+ \right\} + L(\tfrac{h}{L})^{\frac{p-1}{\alpha}}\varrho^p$$

$$\overset{\lambda = \frac{\||\partial f|\|_{\mathbb{P}_n, q}}{p}\varrho^{1-p}}{\leq} \||\partial f|\|_{\mathbb{P}_n, q} \cdot \varrho + L(\tfrac{h}{L})^{\frac{p-1}{\alpha}}\varrho^p + 2L\mathbb{E}_{\mathbb{P}_n} \big[ \big( (\tfrac{Lp}{\||\partial f|\|_{\mathbb{P}_n, q}})^{\frac{1}{p-1}}\varrho - d(\hat{x}^i, \mathscr{D}_f) \big)_+ \big]. \tag{6}$$

If $p \geq 1 + \alpha$, by (4) and (5), we have

$$\sup_{x^i} \{ f(x^i) - f(\hat{x}^i) - \lambda \|x^i - \hat{x}^i\|^p \} \leq \sup_{\delta \geq 0} \{ |\partial f|(\hat{x}^i)\delta - \lambda\delta^p \} + h(\tfrac{L}{\lambda})^{\frac{\alpha+1}{p-1}} + L\big( (\tfrac{L}{\lambda})^{\frac{1}{p-1}} - d(\hat{x}^i, \mathscr{D}_f) \big)_+.$$

It follows that

$$\mathcal{L}_{n,p}^{\mathsf{wo}}(f; \varrho) - \frac{1}{n} \sum_{i=1}^{n} f(\hat{x}^i)$$

$$\leq \min_{\lambda \geq 0} \left\{ \lambda\varrho^p + \frac{1}{n} \sum_{i=1}^{n} \sup_{\delta_i > 0} \{ |\partial f|(\hat{x}^i)\delta_i - \lambda\delta_i^p \} + h(\tfrac{L}{\lambda})^{\frac{\alpha+1}{p-1}} + L\big( (\tfrac{L}{\lambda})^{\frac{1}{p-1}} - d(\hat{x}^i, \mathscr{D}_f) \big)_+ \right\} \tag{7}$$

$$\leq \||\partial f|\|_{\mathbb{P}_n, q} \cdot \varrho + h(\tfrac{Lp}{\||\partial f|\|_{\mathbb{P}_n, q}})^{\frac{\alpha+1}{p-1}}\varrho^{\alpha+1} + 2L\mathbb{E}_{\mathbb{P}_n} \big[ \big( (\tfrac{Lp}{\||\partial f|\|_{\mathbb{P}_n, q}})^{\frac{1}{p-1}}\varrho - d(\hat{x}^i, \mathscr{D}_f) \big)_+ \big],$$

where the last inequality holds by taking a feasible solution $\lambda = \frac{\||\partial f|\|_{\mathbb{P}_n, q}}{p} \varrho^{1-p}$. This completes the proof for the upper bound.

Next, we prove the lower bound. If $p \geq \alpha + 1$, by (4) we have

$$
\frac{1}{n} \sup_{\{x^i\}_i \in \mathscr{X}} \sum_{i=1}^n (f(x^i) - f(\hat{x}^i))
$$

$$
\geq \frac{1}{n} \sum_{i=1}^n \sup_{\delta_i \geq 0} \{ |\partial f|(\hat{x}^i)\delta_i - h\delta_i^{\alpha+1} - 2L(\delta_i - d(\hat{x}^i, \mathscr{D}_f))_+ : \frac{1}{n} \sum_{i=1}^n \delta_i^p \leq \varrho^p \}
$$

$$
\geq \frac{1}{n} \sup_{\delta_i \geq 0} \{ \sum_{i=1}^n |\partial f|(\hat{x}^i)\delta_i - 2L(\delta - d(\hat{x}^i, \mathscr{D}_f))_+ : \frac{1}{n} \sum_{i=1}^n \delta_i^p \leq \varrho^p \} - \frac{1}{n} \sup_{\delta_i \geq 0} \sum_{i=1}^n \{ h\delta_i^{\alpha+1} : \frac{1}{n} \sum_{i=1}^n \delta_i^p \leq \varrho^p \}
$$

$$
\geq \||\partial f|\|_{q, \mathbb{P}_n} \varrho - 2L \cdot \mathbb{E}_{\mathbb{P}_n} \left[ ((\frac{|\partial f|}{\||\partial f|\|_{q, \mathbb{P}_n}})^{q-1} \varrho - d(\hat{x}, \mathscr{D}_f))_+ \right] - h\varrho^{\alpha+1},
$$

where the first term in the last inequality is obtained by taking $\delta_i = \frac{|\partial f|(\hat{x}^i)^{q-1}}{\||\partial f|\|_{q, \mathbb{P}_n}^{q-1}} \varrho$, and the second term is due to Hölder's inequality.

If $p < \alpha + 1$, using Lemma 1 and the fact that the optimal $\delta < (\frac{L}{h})^{\frac{1}{\alpha}}$ (c.f. (5)), we have

$$
\mathcal{L}_{n,p}^{\mathsf{wo}}(f; \varrho) - \frac{1}{n} \sum_{i=1}^n f(\hat{x}^i) \geq \min_{\lambda \geq 0} \left\{ \lambda \varrho^p + \frac{1}{n} \sum_{i=1}^n \sup_{0 \leq \delta \leq (\frac{L}{h})^{\frac{1}{\alpha}}, \|x^i - \hat{x}_i\| \leq \delta} \{f(x_i) - f(\hat{x}^i) - \lambda \delta^p\} \right\},
$$

Using 4 we can bound the inner supremum as

$$
\sup_{0 \leq \delta \leq (\frac{L}{h})^{\frac{1}{\alpha}}, \|x^i - \hat{x}_i\| \leq \delta} \{f(x_i) - f(\hat{x}^i) - \lambda \delta^p\}
$$

$$
\geq \sup_{\delta \geq 0} \{ |\partial f|(\hat{x}^i)\delta - (\lambda + h(\frac{L}{h})^{\frac{\alpha+1-p}{\alpha}})\delta^p - 2L(\delta - d(\hat{x}^i, \mathscr{D}_f))_+ \}.
$$

It follows that

$$
\mathcal{L}_{n,p}^{\mathsf{wo}}(f; \varrho) - \frac{1}{n} \sum_{i=1}^n f(\hat{x}^i)
$$

$$
\geq \min_{\lambda \geq 0} \lambda \varrho^p + \frac{1}{n} \sum_{i=1}^n \sup_{\delta \geq 0} \{ |\partial f|(\hat{x}^i)\delta - (\lambda + h(\frac{L}{h})^{\frac{\alpha+1-p}{\alpha}})\delta^p - 2L(\delta - d(\hat{x}^i, \mathscr{D}_f))_+ \}
$$

$$
\geq \||\partial f|\|_{q, \mathbb{P}_n} \cdot \varrho - h(\frac{L}{h})^{\frac{\alpha+1-p}{\alpha}} \varrho^p - 2L \cdot \mathbb{E}_{\mathbb{P}_n} \left[ ((\frac{|\partial f|(\hat{x})}{\||\partial f|\|_{q, \mathbb{P}_n}})^{q-1} \varrho - d(\hat{x}, \mathscr{D}_f))_+ \right],
$$

where the last inequality holds by taking $\delta = \frac{|\partial f|(\hat{x})^{q-1}}{\||\partial f|\|_{q, \mathbb{P}_n}^{q-1}} \varrho$. Finally, using the assumption that $\||\partial f|\|_{q, \mathbb{P}_n} \geq \tilde{\eta}$ for all $f \in \mathscr{F}$, we get the desired result.

We remark that the result can be extended to arbitrary nominal distributions using exactly the same idea, with summation replaced by integration.

### B.2  Proof of Proposition 2

Since $\mathcal{L}_{n,p}^{\mathsf{wo}}(\varrho) - \mathcal{L}_{n,p}^{\mathsf{ro}}(\varrho) \geq 0$ trivially holds, it suffices to prove the other direction. Fixing $f \in \mathscr{F}$, consider the dual problem from Lemma 1,

$$
\mathcal{L}_{n,p}^{\mathsf{wo}}(\varrho) = \min_{\lambda \geq 0} \left\{ \lambda \varrho^p + \frac{1}{n} \sum_{i=1}^n \sup_{x^i \in \mathscr{X}} \{f(x^i) - \lambda \|x^i - \hat{x}^i\|^p\} \right\}.
$$

Define

$$
v(\lambda) = \lambda \varrho^p + \frac{1}{n} \sum_{i=1}^n \sup_{x \in \mathscr{X}} \{f(x) - \lambda \|x - \hat{x}^i\|^p\},
$$

and let $\lambda_n$ be a minimizer. If $\lambda_n = 0$, then $\mathcal{L}_{n,p}^{\mathsf{wo}}(\varrho) = \mathcal{L}_{n,p}^{\mathsf{ro}}(\varrho)$, since there exists a worst-case distribution that supports on $n$ points according to the structure of the worst-case distribution [19]. In

the sequel we consider $\lambda_n > 0$. By Assumption 1 and the structure of the worst-case distribution [19], $\mathcal{L}_{n,p}^{\text{wo}}(\varrho)$ is attained at a distribution of the form

$$\frac{1}{n}\sum_{i=1}^{n-1} \boldsymbol{\delta}_{x^i} + \frac{1-\epsilon}{n}\boldsymbol{\delta}_{x_-^n} + \frac{\epsilon}{n}\boldsymbol{\delta}_{x_+^n},$$

where

$$x^i \in \arg\max_{x\in\mathscr{X}}\{f(x) - \lambda\|x - \hat{x}^i\|^p\}, \quad i = 1,\ldots,n,$$
$$x_\pm^n \in \arg\max_{x\in\mathscr{X}}\{f(x) - \lambda\|x - \hat{x}^n\|^p\},$$

and

$$\frac{1}{n}\sum_{i=1}^{n-1}\|x^i - \hat{x}^i\|^p + \frac{1-\epsilon}{n}\|x_-^n - \hat{x}^n\|^p + \frac{\epsilon}{n}\|x_+^n - \hat{x}^n\|^p = \varrho^p.$$

By (5), we have that $\|x^i - \hat{x}^i\| \leq (\frac{L}{\lambda_n})^{\frac{1}{p-1}}$. Without loss of generality, we assume that $\|x_+^n - \hat{x}^n\| \geq \|x_-^n - \hat{x}^n\|$, which implies $f(x_+^n) \geq f(x_-^n)$. It follows that $\{x^i, i = 1,\ldots n-1, x_-^n\} \in \mathscr{X}_p(\varrho)$, and we have

$$\max_{\{x^i\}_{i=1}^n \in \mathscr{X}_p(\varrho)} \frac{1}{n}\sum_{i=1}^n f(x^i) \geq \frac{1}{n}\sum_{i=1}^{n-1} f(x^i) + \frac{1}{n}f(x_-^n)$$
$$\geq \mathcal{L}_{n,p}^{\text{wo}}(\varrho) - \frac{\epsilon}{n}(f(x_-^n) - f(x_+^n))$$
$$\geq \mathcal{L}_{n,p}^{\text{wo}}(\varrho) - \frac{2L}{n}(\frac{L}{\lambda_n})^{\frac{1}{p-1}}.$$

It remains to lower bound $\lambda_n$. To this end, observe that by choosing $\lambda_0 = \frac{\||\partial f|\|_{\mathbb{P}_n,q}}{p}\varrho^{-p+1}$, by (6) and (7), we have for all $f \in \mathscr{F}$,

$$v(\lambda_0) - \frac{1}{n}\sum_{i=1}^n f(\hat{x}^i) \leq \||\partial f|\|_{\mathbb{P}_n,q}\cdot\varrho + 2Le_n\left((\frac{Lp}{\||\partial f|\|_{\mathbb{P}_n,q}})^{\frac{1}{p-1}}\varrho\right)$$
$$+ \mathbf{1}\{p < \alpha + 1\}L(\frac{h}{L})^{\frac{p-1}{\alpha}}\varrho^p + \mathbf{1}\{p \geq \alpha + 1\}h(\frac{Lp}{\||\partial f|\|_{\mathbb{P}_n,q}})\varrho^{\alpha+1},$$

recalling that $e_n$ is defined in (2). Also note that

$$v(\lambda_n) - \frac{1}{n}\sum_{i=1}^n f(\hat{x}^i) \overset{(4)}{\geq} \lambda_n\varrho^p + \frac{1}{n}\sum_{i=1}^n \sup_{\delta_i\geq 0}\{|\partial f|(\hat{x}^i)\delta_i - h\delta_i^{\alpha+1} - 2L(\delta_i - d(\hat{x}^i,\mathscr{D}_f))_+ - \lambda_n\delta_i^p\}$$
$$\overset{\delta_i=2\varrho}{\geq} 2\||\partial f|\|_{q,\mathbb{P}_n}\cdot\varrho - 2^{\alpha+1}\varrho^{\alpha+1}h - (2^p - 1)\varrho^p\lambda_n - 2Le_n(2\varrho).$$

If

$$(2^p - 1)\varrho^p\lambda_n < \||\partial f|\|_{q,\mathbb{P}_n}\cdot\varrho - 2^{\alpha+1}\varrho^{\alpha+1}h - 2Le_n(2\varrho)$$
$$- 2Le_n\left((\frac{Lp}{\||\partial f|\|_{\mathbb{P}_n,q}})^{\frac{1}{p-1}}\varrho\right) - L(\frac{h}{L})^{\frac{p-1}{\alpha}}\varrho^p - h(\frac{Lp}{\||\partial f|\|_{\mathbb{P}_n,q}})\varrho^{\alpha+1},$$

then

$$v(\lambda_n) - \frac{1}{n}\sum_{i=1}^n f(\hat{x}^i)$$
$$> \||\partial f|\|_{q,\mathbb{P}_n}\cdot\varrho + 2Le_n\left((\frac{Lp}{\||\partial f|\|_{\mathbb{P}_n,q}})^{\frac{1}{p-1}}\varrho\right) + L(\frac{h}{L})^{\frac{p-1}{\alpha}}\varrho^p + h(\frac{Lp}{\||\partial f|\|_{\mathbb{P}_n,q}})\varrho^{\alpha+1}$$
$$\geq v(\lambda_0) - \frac{1}{n}\sum_{i=1}^n f(\hat{x}^i),$$

which is contradicted to the optimality of $\lambda_n$. Hence, we must have

$$(2^p - 1)\varrho^p\lambda_n \geq \||\partial f|\|_{q,\mathbb{P}_n}\cdot\varrho - 2^{\alpha+1}\varrho^{\alpha+1}h - 2Le_n(2\varrho)$$
$$- 2Le_n\left((\frac{Lp}{\||\partial f|\|_{\mathbb{P}_n,q}})^{\frac{1}{p-1}}\varrho\right) - L(\frac{h}{L})^{\frac{p-1}{\alpha}}\varrho^p - h(\frac{Lp}{\||\partial f|\|_{\mathbb{P}_n,q}})\varrho^{\alpha+1},$$

hence we complete the proof.

## B.3 Proof of Proposition 3

In the sequel, when the loss function $f \in \mathscr{F}$ is clear from the context, we denote $H(x), G(x)$ for $H_f(x), G_f(x)$, and $G_i$ for $G(\hat{x}_i)$ to simplify the notation. The upper bound is straightforward.

To prove the lower bound, we first consider (WO), without loss of generality, we assume that $G(\hat{x}^1) \geq G(\hat{x}^2) \geq \ldots G(\hat{x}^n)$, otherwise we just relabel them. Suppose the maximum in defining $G_i$ is attained at $\tilde{x}^i$ with distance $d_i$ from $\hat{x}^i$; otherwise we can use an approximation argument. Starting from $i = 1$, if $d_i > n^\delta \varrho$, we set $x^i = \tilde{x}^i$, otherwise we set $x^i = \hat{x}^i + n^\delta \varrho u_i$, where $u_i$ is a direction to be determined later, i.e., we perturb $\hat{x}^i$ with distance $\delta_i = \|x^i - \hat{x}^i\| = \max(d_i, n^\delta \varrho)$, in the direction of $\tilde{x}^i - \hat{x}^i$ if $d_i > n^\delta \varrho$, or $u_i$ otherwise. Then we proceed to $i + 1$, until we achieve the largest $N$ such that $\frac{1}{n} \sum_{i=1}^N \delta_i \leq \varrho$. Note that since each $\delta_i \geq n^\delta \varrho$, we have $N \leq \lfloor n^{1-\delta} \rfloor$. If for some $i$ we cannot perturb it with fully mass $1/n$, we perturb $\frac{\epsilon}{n}$ of $\hat{x}_{N+1}$, so that $\frac{1}{n} \sum_{i=1}^N \delta_i + \frac{\epsilon}{n} \delta_{N+1} = \varrho$, where $0 \leq \epsilon \leq 1$. By construction, $\{x^i\}_{i=1\ldots n} \in \mathscr{X}_1(\varrho)$. Hence we have

$$\frac{1}{n} \sum_{i=1}^n (f(x^i) - f(\hat{x}^i))$$

$$\geq \frac{1}{n} \sum_{i=1}^N (f(x^i) - f(\hat{x}^i)) + \frac{\epsilon}{n}(f(x^{N+1}) - f(\hat{x}^{N+1}))$$

$$= \frac{1}{n} \Big[ \sum_{i:d_i > n^\delta \varrho} (f(x^i) - f(\hat{x}^i)) + \sum_{i:d_i \leq n^\delta \varrho} (f(x^i) - f(\hat{x}^i)) \Big] + \frac{\epsilon}{n}(f(x^{N+1}) - f(\hat{x}^{N+1}))$$

$$= \frac{1}{n} \Big[ \sum_{i:d_i > n^\delta \varrho} \delta_i G_i + \sum_{i:d_i \leq n^\delta \varrho} (f(x^i) - f(\hat{x}^i)) \Big] + \frac{\epsilon}{n}(f(x^{N+1}) - f(\hat{x}^{N+1})).$$

To bound second sum above involving $d_i \leq n^\delta \varrho$, by (4), we choose $u_i$ (and thus $x^i$) so that

$$f(x^i) - f(\hat{x}^i) \geq |\partial f|(\hat{x}^i)\delta_i - \left(h \cdot \delta_i^{\alpha+1} + L\big(\delta_i - d(\hat{x}^i, \mathscr{D}_f)\big)_+\right). \tag{8}$$

On the other hand, to bound the first sum above involving the difference between $G_i$ and $|\partial f|(\hat{x}^i)$, by (4) and $\delta_i \geq d_i > 0$ when $d_i \leq n^\delta \varrho$, we have

$$G_i d_i = f(\tilde{x}^i) - f(\hat{x}^i) \leq |\partial f|(\hat{x}^i)d_i + h \cdot d_i^{\alpha+1} + L\big(d_i - d(\hat{x}^i, \mathscr{D}_f)\big)_+.$$

Hence,

$$\begin{aligned}
(G_i - |\partial f|(\hat{x}^i))\delta_i &= (G_i - |\partial f|(\hat{x}^i))d_i \cdot \frac{\delta_i}{d_i} \\
&\leq h \cdot d_i^\alpha \delta_i + L\big(d_i - d(\hat{x}^i, \mathscr{D}_f)\big)_+ \cdot \frac{\delta_i}{d_i} \\
&\leq h \cdot \delta_i^{\alpha+1} + L\big(\delta_i - d(\hat{x}^i, \mathscr{D}_f)\big)_+.
\end{aligned} \tag{9}$$

Therefore we have

$$\frac{1}{n} \sum_{i=1}^n (f(x^i) - f(\hat{x}^i))$$

$$\geq \frac{1}{n} \Big[ \sum_{i:d_i > n^\delta \varrho} \delta_i G_i + \sum_{i:d_i \leq n^\delta \varrho} G_i \delta_i - 2\big(h \cdot \delta_i^{\alpha+1} + L\big(\delta_i - d(\hat{x}^i, \mathscr{D}_f)\big)_+\big) \Big] + \frac{\epsilon}{n} \delta_{N+1} G_{N+1}$$

$$\geq \frac{1}{n} \sum_{i=1}^N \delta_i G_i + \frac{\epsilon}{n} \delta_{N+1} G_{N+1} - \frac{2}{n} \sum_{i=1}^{N+1} \big(h \cdot (n^\delta \varrho)^{\alpha+1} + L\big((n^\delta \varrho) - d(\hat{x}^i, \mathscr{D}_f)\big)_+\big)$$

$$\geq \varrho(\|f\|_{\text{Lip}} - \Delta_n) - \frac{2h(N+1)}{n}(n^\delta \varrho)^{\alpha+1} - 2L\mathbb{E}_{\mathbb{P}_n}\big[\big(n^\delta \varrho - d(x, \mathscr{D}_f)\big)_+\big]$$

$$\geq \varrho(\|f\|_{\text{Lip}} - \Delta_n) - (4h\varrho^{\alpha+1})n^{\delta\alpha} - 2Le_n(n^\delta \varrho),$$

Next we consider (RO). Fix $N = \lfloor n^{1-\delta} \rfloor + 1$, and we just perturb the first $\lfloor n^{1-\delta} \rfloor$ points $x^i$ with distance $\delta_i = \|x^i - \hat{x}^i\| = \frac{n\varrho}{N} \leq n^\delta \varrho$ in the direction $u^i$ so that

$$f(x^i) - f(\hat{x}^i) \geq |\partial f|(\hat{x}^i)\delta_i - \left(h \cdot \delta_i^{\alpha+1} + L\big(\delta_i - d(\hat{x}^i, \mathscr{D}_f)\big)_+\right),$$

and remain the rest points unchanged. Then,

$$\tfrac{1}{n}\sum_{i=1}^{n}\big(f(x^i)-f(\hat{x}^i)\big)$$

$$\geq \tfrac{1}{n}\sum_{i=1}^{n}\left[|\partial f|(\hat{x}^i)\delta_i - \big(h\cdot\delta_i^{\alpha+1} + L\big(\delta_i - d(\hat{x}^i,\mathscr{D}_f)\big)_+\big)\right]$$

$$\geq \frac{1}{n}\sum_{i=1}^{N}\delta_i|\partial f|(\hat{x}^i) - \frac{1}{n}\sum_{i=1}^{N}\big(h\cdot(n^\delta\varrho)^{\alpha+1} + L\big((n^\delta\varrho) - d(\hat{x}^i,\mathscr{D}_f)\big)_+\big)$$

$$\geq |\partial f|(\hat{x}^{N+1})\varrho - \frac{hN}{n}(n^\delta\varrho)^{\alpha+1} - L\mathbb{E}_{\mathbb{P}_n}\big[\big(n^\delta\varrho - d(x,\mathscr{D}_f)\big)_+\big]$$

$$\geq |\partial f|(\hat{x}^{N+1})\varrho - (2h\varrho^{\alpha+1})n^{\delta\alpha} - L\mathbb{E}_{\mathbb{P}_n}\big[\big(n^\delta\varrho - d(x,\mathscr{D}_f)\big)_+\big].$$

The rest follows a similar argument as for (WO). □

*Proof of Remark 2.* Here we bound $\Delta_n$. Observe that $H_f(\Delta_n)$ is the $(\lfloor n^{1-\delta}\rfloor + 1)$-th order statistic of $n$ i.i.d samplings from the unit uniform distribution, which follows Beta distribution $B(\lfloor n^{1-\delta}\rfloor + 1, n - \lfloor n^{1-\delta}\rfloor)$, which is sub-Gaussian with proxy variance $\frac{1}{4(n+1)}$ [28]. Hence, for any $t > 0$, with probability at least $1 - e^{-2nt^2}$, we have

$$H(\Delta_n) - \frac{\lfloor n^{1-\delta}\rfloor + 1}{n+1} < t.$$

Replacing $t$ with $\frac{t}{\sqrt{n}}$, we have with probability at least $1 - e^{-2t}$,

$$\Delta_n \leq H^{-1}\Big(\frac{t}{\sqrt{n}} + \frac{\lfloor n^{1-\delta}\rfloor + 1}{n+1}\Big) \leq c^{-\beta}\Big(\frac{t}{\sqrt{n}} + \frac{n^{1-\delta}+1}{n+1}\Big)^\beta,$$

whenever $\frac{t}{\sqrt{n}} + \frac{\lfloor n^{1-\delta}\rfloor+1}{n+1} \leq \bar{a}$. When $\varrho = \varrho_n = O(1/\sqrt{n})$, by [18, Theorem 1] we have $\varrho_n\Delta_n + 4hn^{\alpha\delta}\varrho^{\alpha+1} + 2Le_n(\varrho n^\delta) = O(n^{-\frac{\beta+1}{2}} + n^{-(\frac{1}{2}+\beta\delta)} + n^{-(1-2\delta)})$. □

## C  Proofs for Section 4

### C.1  Proofs of Propositions 4 and 5

In the sequel, we set $\mathbf{x}^n := (x^1,\ldots,x^n)$, and define a metric $\mathbf{d}$ on $\mathscr{X}^n$ as $\mathbf{d}(\tilde{\mathbf{x}}^n,\mathbf{x}^n) = (\sum_{i=1}^n\|\tilde{x}^i - x^i\|^p)^{1/p}$. The following result follows from the proof of Lemma 5 in [17] by replacing $\mathbb{E}_\otimes$ and $\tau$ therein with $\mathbb{E}_{S_n}$ and $\tau_n n^{1-\frac{2}{p}}$, respectively.

**Lemma 6.** *Let $p \in [1,2]$. Assume Assumption 5 holds. Let $F : \mathscr{X}^n \to \mathbb{R}$. Assume $\mathbb{E}_{S_n}[F] = 0$ and there exist $M, L > 0$ and $\mathbf{x}_0^n \in \mathscr{X}^n$ such that*

$$F(\mathbf{x}^n) \leq M + \frac{L}{n}\mathbf{d}(\mathbf{x}^n,\mathbf{x}_0^n)^p, \ \ \forall \mathbf{x}^n \in \mathscr{X}^n.$$

*Define $\mathcal{R}(\cdot;F) : \mathbb{R}_+ \to \mathbb{R}_+$ as*

$$\mathcal{R}_{S_n,p}(\varrho;F) = \min_{\lambda\geq 0}\left\{\lambda\varrho^p + \mathbb{E}_{S_n}\left[\sup_{\tilde{\mathbf{x}}^n\in\mathscr{X}^n}\left\{F(\tilde{\mathbf{x}}^n) - F(\mathbf{x}^n) - \frac{\lambda}{n}\mathbf{d}(\tilde{\mathbf{x}}^n,\mathbf{x}^n)^p\right\}\right]\right\}.$$

*Let $t > 0$. Then with probability at least $1 - e^{-t}$,*

$$F(\mathbf{x}^n) \leq \mathcal{R}_{S_n,p}\left(\sqrt{\frac{\tau_n t}{n^{\frac{2}{p}}}}; F\right).$$

*Proof of Propositions 4 and 5.* We first fix $f \in \mathscr{F}$. Set $F(\mathbf{x}^n) = \mathbb{E}_{\mathbb{P}_{\text{true}}}[f] - \mathbb{E}_{\mathbb{P}_n}[f]$. By definition $\mathbb{E}_{S_n}[F(\mathbf{x}^n)] = 0$. By Assumption 1 we have

$$F(\tilde{\mathbf{x}}^n) - F(\mathbf{x}^n) \leq \sup_{f \in \mathscr{F}} \left\{ \frac{1}{n} \sum_{i=1}^n f(\tilde{x}^i) - f(x^i) \right\}$$

$$\leq \frac{1}{n} \sum_{i=1}^n L \|\tilde{x}^i - x^i\|$$

$$\leq \frac{1}{n} \sum_{i=1}^n L(1 + \|\tilde{x}^i - x^i\|^p)$$

$$= L + L\mathbf{d}(\tilde{\mathbf{x}}^n, \mathbf{x}^n)^p.$$

Hence, by Lemma 6, with probability at least $1 - e^{-t}$, we have

$$\mathbb{E}_{\mathbb{P}_{\text{true}}}[f] - \mathbb{E}_{\mathbb{P}_n}[f]$$
$$\leq F(\mathbf{x}^n)$$
$$\leq \mathcal{R}_{S_n, p}\left( \sqrt{\frac{\tau_n t}{n^{\frac{2}{p}}}}; F \right)$$
$$= \min_{\lambda \geq 0} \left\{ \lambda \left( \sqrt{\frac{\tau_n t}{n^{\frac{2}{p}}}} \right)^p + \mathbb{E}_{S_n} \left[ \sup_{\tilde{\mathbf{x}}^n \in \mathscr{X}^n} \left\{ F(\tilde{\mathbf{x}}^n) - F(\mathbf{x}^n) - \frac{\lambda}{n} \mathbf{d}(\tilde{\mathbf{x}}^n, \mathbf{x}^n)^p \right\} \right] \right\}$$
$$\leq \min_{\lambda \geq 0} \left\{ \lambda \left( \sqrt{\frac{\tau_n t}{n^{\frac{2}{p}}}} \right)^p + \mathbb{E}_{S_n} \left[ \frac{1}{n} \sup_{\tilde{\mathbf{x}}^n \in \mathscr{X}^n} \sum_{i=1}^n \left( -f(\tilde{x}^i) + f(x^i) - \lambda \|\tilde{x}^i - x^i\|^p \right) \right] \right\}$$
$$= \min_{\lambda \geq 0} \left\{ \lambda \left( \sqrt{\frac{\tau_n t}{n^{\frac{2}{p}}}} \right)^p + \mathbb{E}_{\mathbb{P}_{\text{true}}} \left[ \sup_{\tilde{x} \in \mathscr{X}} \left\{ -f(\tilde{x}) + f(x) - \lambda \|\tilde{x} - x\|^p \right\} \right] \right\}.$$

We denote the last line as $\mathcal{R}_p \left( \sqrt{\frac{\tau_n t}{n^{\frac{2}{p}}}}; -f \right)$. Note that Assumption 1 implies that for any $\lambda > \|f\|_{\text{Lip}}$,

$$\sup_{\tilde{x} \in \mathscr{X}} \left\{ -f(\tilde{x}) - f(x) - \lambda \|\tilde{x} - x\| \right\} = 0.$$

Consequently by definition $\mathcal{R}_1(\varrho; -f) \leq \varrho \|f\|_{\text{Lip}}$ for all $\varrho \geq 0$. Moreover, for $p \in (1, 2]$, by Proposition 1 (in which $\mathbb{P}_n$ is replaced with $\mathbb{P}_{\text{true}}$), we have

$$\mathcal{R}_p(\varrho; -f) \leq \varrho \|\|\partial(-f)\|\|_q + C_0 \varrho^{(\alpha+1) \wedge p} + 2Le\big( (pL/\eta)^{\frac{1}{p-1}} \varrho \big)$$
$$\leq \left( 1 - 2\mathfrak{R}_n(\mathcal{N}_q) - (L/\eta)^q \sqrt{\frac{t}{2n}} \right)^{-\frac{1}{q}} \varrho \|\|\partial(-f)\|\|_{q, \mathbb{P}_n} + C_0 \varrho^{(\alpha+1) \wedge p} + 2Le\big( (pL/\eta)^{\frac{1}{p-1}} \varrho \big),$$

with probability at least $1 - e^{-t}$, following Lemma 5.

To obtain a uniform bound, by Assumption 6, for any distribution $\mathbb{P}$ it holds that

$$\mathbb{E}_{\mathbb{P}}[f_{\theta'}] - \mathbb{E}_{\mathbb{P}}[f_\theta] \leq \kappa \|\theta' - \theta\|_\Theta.$$

Let $\epsilon > 0$ and $\Theta_\epsilon$ be an $\epsilon$-cover of $\Theta$.

When $p = 1$, we have that

$$\mathbb{P}_{\mu_{S_n}} \left\{ \exists \theta \in \Theta, s.t. \mathcal{L}^{\text{true}}(f_\theta) > \mathcal{L}_{n,\infty}^{\text{vr}}(f_\theta; \varrho) + 2\kappa\epsilon \right\}$$
$$= \mathbb{P}_{\mu_{S_n}} \left\{ \exists \theta \in \Theta, s.t. \mathbb{E}_{\mathbb{P}_{\text{true}}}[f_\theta] > \mathbb{E}_{\mathbb{P}_n}[f_\theta] + \varrho \|f_\theta\|_{\text{Lip}} + 2\kappa\epsilon \right\}$$
$$\leq \mathbb{P}_{\mu_{S_n}} \left\{ \exists \theta' \in \Theta_\epsilon, s.t. \mathbb{E}_{\mathbb{P}_{\text{true}}}[f_{\theta'}] > \mathbb{E}_{\mathbb{P}_n}[f_{\theta'}] + \varrho \|f_\theta\|_{\text{Lip}} \right\}$$
$$\leq \sum_{\theta' \in \Theta_\epsilon} \mathbb{P}_{\mu_{S_n}} \left\{ \mathbb{E}_{\mathbb{P}_{\text{true}}}[f_{\theta'}] > \mathbb{E}_{\mathbb{P}_n}[f_{\theta'}] + \varrho \|f_\theta\|_{\text{Lip}} \right\}.$$

Letting $\epsilon = 1/n$ yields that with probability at least $1 - \mathscr{N}(\epsilon; \Theta, \|\cdot\|_\Theta)e^{-t}$, for every $\theta \in \Theta$,

$$\mathbb{E}_{\mathbb{P}_{\mathsf{true}}}[f_\theta] - \mathbb{E}_{\mathbb{P}_n}[f_\theta] \le \mathcal{R}_1\Big(\sqrt{\frac{\tau_n t}{n^2}}; -f_\theta\Big) + \frac{2\kappa}{n}.$$

Replacing $t$ with $t + \log \mathscr{N}(\epsilon; \Theta, \|\cdot\|_\Theta)$ yields Proposition 4.

When $p \in (1, 2]$, let

$$\tilde{\epsilon} = C_0 \varrho^{(\alpha+1)\wedge p} + 2Le\big((pL/\eta)^{\frac{1}{p-1}}\varrho\big).$$

We have that

$$\mathbb{P}_{\mu_{S_n}}\Big\{\exists \theta \in \Theta, s.t.\ \mathcal{L}^{\mathsf{true}}(f_\theta) > \mathcal{L}_{n,q}^{\mathsf{vr}}(f_\theta; \varrho) + 2\kappa\epsilon + \tilde{\epsilon}\Big\}$$

$$= \mathbb{P}_{\mu_{S_n}}\Big\{\exists \theta \in \Theta, s.t.\ \mathbb{E}_{\mathbb{P}_{\mathsf{true}}}[f_\theta] > \mathbb{E}_{\mathbb{P}_n}[f_\theta] + \varrho\|\|\partial f_\theta\|\|_q + 2\kappa\epsilon + \tilde{\epsilon}\Big\}$$

$$= \mathbb{P}_{\mu_{S_n}}\Big\{\exists \theta \in \Theta, s.t.\ \mathbb{E}_{\mathbb{P}_{\mathsf{true}}}[f_\theta] > \mathbb{E}_{\mathbb{P}_n}[f_\theta] + \varrho\|\|\partial(-f_\theta)\|\|_q + 2\kappa\epsilon + \tilde{\epsilon}\Big\}$$

$$\le e^{-t} + \mathbb{P}_{\mu_{S_n}}\Big\{\exists \theta' \in \Theta_\epsilon, s.t.\ \mathbb{E}_{\mathbb{P}_{\mathsf{true}}}[f_{\theta'}] > \mathbb{E}_{\mathbb{P}_n}[f_{\theta'}] + \big(1 - 2\mathfrak{R}_n(\mathscr{N}_q) - (L/\eta)^q\sqrt{\tfrac{t}{2n}}\big)^{-\frac{1}{q}}\varrho\|\|\partial(-f_{\theta'})\|\|_{q,\mathbb{P}_n} + \tilde{\epsilon}\Big\}$$

$$\le e^{-t} + \sum_{\theta' \in \Theta_\epsilon} \mathbb{P}_{\mu_{S_n}}\Big\{\mathbb{E}_{\mathbb{P}_{\mathsf{true}}}[f_{\theta'}] > \mathbb{E}_{\mathbb{P}_n}[f_{\theta'}] + \big(1 - 2\mathfrak{R}_n(\mathscr{N}_q) - (L/\eta)^q\sqrt{\tfrac{t}{2n}}\big)^{-\frac{1}{q}}\varrho\|\|\partial(-f_{\theta'})\|\|_{q,\mathbb{P}_n} + \tilde{\epsilon}\Big\}.$$

Letting $\epsilon = 1/n$ yields that with probability at least $1 - (1 + \mathscr{N}(\tfrac{1}{n}; \Theta, \|\cdot\|_\Theta))e^{-t}$, for every $\theta \in \Theta$,

$$\mathbb{E}_{\mathbb{P}_{\mathsf{true}}}[f_\theta] - \mathbb{E}_{\mathbb{P}_n}[f_\theta] \le \mathcal{L}_{n,q}^{\mathsf{vr}}(f_\theta; \varrho_n) + \frac{2\kappa}{n} + \tilde{\epsilon}_n.$$

Finally, replacing $t$ with $t + \log(1 + \mathscr{N}(\tfrac{1}{n}; \Theta, \|\cdot\|_\Theta))$ yields Proposition 5. $\qquad\square$

## C.2 Proof for Example 1

**Lemma 7.** *We gather a few simple facts about Assumption 6:*

- (i) *If $\{f_\theta : \theta \in \Theta\}$ satisfies Assumption 6 with parameter $\kappa$, and $\phi$ is a L-Lipschitz over the range of all functions in $\{f_\theta : \theta \in \Theta\}$, then $\{\phi \circ f_\theta : \theta \in \Theta\}$ satisfies Assumption 6 with parameter $L\kappa$.*

- (ii) *If both $\{f_\theta : \theta \in \Theta\}$ and $\{g_\theta : \theta \in \Theta\}$ satisfy Assumption 6 with parameters $\kappa_1$ and $\kappa_2$, then $\{af_\theta + bg_\theta : \theta \in \Theta\}$ satisfies Assumption 6 with parameters $a\kappa_1 + b\kappa_2$ for any constants $a, b \ge 0$.*

- (iii) *$\{x \mapsto \theta^\top x : \theta \in \Theta, \|x\|_2 \le B\}$ satisfies Assumption 6 with parameter $B$ when $\|\theta\| = \|\cdot\|_2$.*

- (iv) *$\{x \mapsto \|W^\top x\|_2 : W \in \mathcal{W} \subset \mathbb{R}^{d \times k}, \|x\|_2 \le B\}$ satisfies Assumption 6 with parameter $B$ when $\|W\|_\mathcal{W} = \|W\|_F$.*

*Proof.* (i) $|\phi \circ f_{\theta_1} - \phi \circ f_{\theta_2}| \le L|f_{\theta_1} - f_{\theta_2}| \le L\kappa\|\theta_1 - \theta_2\|$.

(ii) $|af_{\theta_1} + bg_{\theta_1} - af_{\theta_2} - bg_{\theta_2}| \le a|f_{\theta_1} - f_{\theta_2}| + b|g_{\theta_1} - g_{\theta_2}| \le (a\kappa_1 + b\kappa_2)\|\theta_1 - \theta_2\|_2$.

(iii) $|\theta_1^\top x - \theta_2^\top x| \le \|\theta_1 - \theta_2\|_2 \|x\|_2 \le B\|\theta_1 - \theta_2\|_2$.

(iv) $|\|W_1^\top x\|_2 - \|W_2^\top x\|_2| \le \|(W_1 - W_2)^\top x\|_2 \le \|W_1 - W_2\|_2\|x\|_2 \le B\|W_1 - W_2\|_2$.

$\qquad\square$

Set $\tilde{\theta} = [\theta, -1]$, then $f_{\tilde{\theta}}(x, y) = |\tilde{\theta}^\top(x, y)|^p \in \mathcal{F}$. It is clear from the definition that every $f \in \mathcal{F}$ is piece-wise differentiable. We assumed both feature space $\mathscr{X}$ and the weight space $\Theta$ are bounded: $\|x\|_2 \le B_1$ for all $x \in \mathscr{X}$, $|y| \le B_2$ for all $y \in \mathscr{Y}$, and $\|\theta\|_2 \le B_3 - 1$ for all $\theta \in \Theta$. Note that $|\cdot|^p$ is Lipschitz, with constant bounded by the upper bound of the gradient norm:

$$p|\tilde{\theta}^\top(x, y)|^{p-1} \le p\|\tilde{\theta}\|_2^{p-1}\|(x, y)\|_2^{p-1} \le pB_3^{p-1}(B_1 + B_2)^{p-1}, \qquad (10)$$

hence Assumption 1 is verified.

To verify Assumption 6, observe that by (10), $|\cdot|^p$ is Lipschitz over the range of $\tilde{\theta}^\top(x, y)$, for all $\theta \in \Theta$. The verification follows from Lemma 7 (i) and (iii).

# D  Proofs for Section 5

## D.1  Proofs for Example 3

Let us verify $\mathscr{F}$ is a family of Lipschitz functions, hence Assumption 1.

$$|f_W(x) - f_W(\tilde{x})| \leq \|W^\top(x - \tilde{x})\|_2 (\|W^\top\tilde{x}\|_2 + \|W^\top x\|_2)$$
$$\leq 2B\|W\|_F^2\|x - \tilde{x}\|_2,$$

hence we get that $f_W$ is $2Bk$-Lipschitz.

Moreover, we have that

$$\mathbb{E}_{\mathbb{P}_{\text{true}}}[\|\nabla f_W\|_2^2] = \mathbb{E}_{\mathbb{P}_{\text{true}}}[\|2WW^\top x\|_2^2]$$
$$= 4\mathbb{E}_{\mathbb{P}_{\text{true}}}[x^\top WW^\top x]$$
$$= 4\text{Tr}(W^\top \mathbb{E}_{\mathbb{P}_{\text{true}}}[xx^\top]W)$$
$$= \sum_{j=1}^k w_j^\top \mathbb{E}_{\mathbb{P}_{\text{true}}}[xx^\top]w_j$$
$$\geq 4k\lambda_{\min}\mathbb{E}_{\mathbb{P}_{\text{true}}}[xx^\top],$$

hence Assumption 3 is verified with $\eta = 2\sqrt{k\lambda_{\min}\mathbb{E}_{\mathbb{P}_{\text{true}}}[xx^\top]}$.

To verify Assumption 6, observe that $\|W^\top x\|_2 \leq \|W\|_F\|x\|_2 \leq \sqrt{k}B$, hence $(\cdot)^2$ is Lipschitz over the range of $\|W^\top x\|_2$, where $W^\top W = I_k$. Hence, using Lemma 7(i)(iv) we have $\kappa = 2\sqrt{k}B^2$.

## D.2  Proofs for Section 5.2

**Lemma 8.** *Assume Assumptions 1, 2 and 4 hold, and $\mathbb{P}_{\text{true}}$ is continuous, and $\varrho \leq c^{\frac{1}{\delta}}(\frac{\bar{a}}{2})^{\frac{1}{\beta\delta}}$, where the constants are from Assumption 4. Then*

$$|\mathcal{L}_1^{\text{adv}}(\varrho; f) - \mathcal{L}_\infty^{\text{vr}}(\varrho; f)| \leq C\varrho^{(1 + \frac{\alpha\beta}{\alpha+\beta})}, \quad \forall f \in \mathscr{F}.$$

*Proof.* For every $x \in \mathscr{X}$, let $S(x)$ be such that

$$f(S(x)) - f(x) = d(x, S(x))G(f)(x),$$

where we have assumed the existence of the maximizer defining $G(f)(x)$, otherwise we can argue by approximation. Let $\epsilon, \delta > 0$. Set $\mathscr{X}_\epsilon = \{x \in \mathscr{X} : G(f)(x) > \|G(f)\|_{\infty, \mathbb{P}_{\text{true}}} - \epsilon\}$.

Define a mapping $T_\epsilon : \mathscr{X} \to \mathscr{X}$ as

$$T_\epsilon(x) = \begin{cases} S(x), & \text{if } x \in \mathscr{X}_\epsilon, \, d(x, S(x)) \geq \varrho^{1-\delta}, \\ x + \varrho^{1-\delta}u, & \text{if } x \in \mathscr{X}_\epsilon, \, d(x, S(x)) < \varrho^{1-\delta}, \\ x, & \text{if } x \in (\mathscr{X} \setminus \mathscr{X}_\epsilon) \cup \mathscr{D}_f, \end{cases}$$

where $u$ is the direction such that

$$f(T_\epsilon(x)) - f(x) \geq |\partial f|(x) \cdot d(x, T_\epsilon(x)) - h \cdot d(x, T_\epsilon(x))^{\alpha+1},$$

which holds due to (4).

Define the monotonically increasing function

$$M(\varepsilon) = \mathbb{E}_{\mathbb{P}_{\text{true}}}[d(x, T_\varepsilon(x))\mathbf{1}\{x \in \mathscr{X}_\varepsilon, \, d(x, S(x)) \geq \varrho^{1-\delta}\}] + \varrho^{1-\delta}\mathbb{E}_{\mathbb{P}_{\text{true}}}[\mathbf{1}\{x \in \mathscr{X}_\varepsilon, \, d(x, S(x)) < \varrho^{1-\delta}\}],$$

and define

$$\epsilon = \inf_{\varepsilon \geq 0}\{M(\varepsilon) \geq \varrho\}.$$

By Assumption 4, we have

$$M(\frac{\bar{a}}{2}) \geq \varrho^{1-\delta}\mathbb{E}_{\mathbb{P}_{\text{true}}}[\mathbf{1}\{x \in \mathscr{X}_{\frac{\bar{a}}{2}}\}] \geq \varrho^{1-\delta}c(\frac{\bar{a}}{2})^{\frac{1}{\beta}} \geq \varrho.$$

It follows that $\epsilon \le \frac{\bar{a}}{2} < \bar{a}$. For any $\epsilon_0 < \epsilon$, we similarly have

$$\varrho > M(\epsilon_0) \ge \varrho^{1-\delta}\mathbb{E}_{\mathbb{P}_{\text{true}}}[\mathbf{1}\{x \in \mathscr{X}_{\epsilon_0}\}] \ge \varrho^{1-\delta}c(\epsilon_0)^{\frac{1}{\beta}}.$$

Taking the limit $\epsilon_0 \to \epsilon$, we get that $\varrho^{1-\delta}c\epsilon^{\frac{1}{\beta}} \le \varrho$, which implies $\epsilon \le (\frac{\varrho^{\delta}}{c})^{\beta}$. Define $\epsilon_1 = \epsilon + (\frac{\varrho^{\delta}}{c})^{\beta}$. Since $M(\epsilon_1) \ge \varrho$, we choose $r \le 1$ such that $rM(\epsilon_1) = \varrho$. Now we define a distribution $\mathbb{P} = (1-r)\mathbb{P}_{\text{true}} + rT_{\epsilon_1\#}\mathbb{P}_{\text{true}}$. Then

$$\mathbb{E}_{\mathbb{P}}[f(x)] - \mathbb{E}_{\mathbb{P}_{\text{true}}}[f(x)] = r\big(\mathbb{E}_{\mathbb{P}_{\text{true}}}[f(T_{\epsilon_1}(x))] - \mathbb{E}_{\mathbb{P}_{\text{true}}}[f(x)]\big).$$

For $x \in \mathscr{X}_{\epsilon_1}$, if $d(x, S(x)) \ge \varrho^{1-\delta}$, we have $f(T_{\epsilon_1}(x)) - f(x) = G(f)(x) \cdot d(x, T_{\epsilon_1}(x))$, and if $d(x, S(x)) < \varrho^{1-\delta}$, similar to (8) (9) in proof of Proposition 3, it holds that

$$f(T_{\epsilon_1}(x)) - f(x) \ge G(f)(x)d(T_{\epsilon_1}(x), x) - 2h \cdot d(T_{\epsilon_1}(x), x)^{\alpha+1}.$$

It follows that

$$\begin{aligned}
\mathbb{E}_{\mathbb{P}}[f(x)] - \mathbb{E}_{\mathbb{P}_{\text{true}}}[f(x)] &= r\big(\mathbb{E}_{\mathbb{P}_{\text{true}}}[f(T_{\epsilon_1}(x))] - \mathbb{E}_{\mathbb{P}_{\text{true}}}[f(x)]\big) \\
&\ge rM(\varepsilon_1)(\|G(f)\|_{\infty,\mathbb{P}_{\text{true}}} - \epsilon_1) - 2rh\varrho^{(1-\delta)(\alpha+1)}\mathbb{E}_{\mathbb{P}_{\text{true}}}[\mathbf{1}\{x \in \mathscr{X}_{\epsilon_1}\}] \\
&\ge \varrho\|G(f)\|_{\infty,\mathbb{P}_{\text{true}}} - 2c^{-\beta}\varrho^{1+\delta\beta} - 2h\varrho^{(1-\delta)(\alpha+1)+\delta},
\end{aligned}$$

where the last row is due to $r\mathbb{E}_{\mathbb{P}_{\text{true}}}[\mathbf{1}\{x \in \mathscr{X}_{\epsilon_1}\}]\varrho^{1-\delta} \le rM(\epsilon_1) = \varrho$.

Setting $\delta = \frac{\alpha}{\alpha+\beta}$ yields that

$$\mathbb{E}_{\mathbb{P}_{\text{true}}}[f(T_{\epsilon_1}(x))] - \mathbb{E}_{\mathbb{P}_{\text{true}}}[f(x)] \ge \varrho\|G(f)\|_{\infty,\mathbb{P}_{\text{true}}} - C\varrho^{(1+\frac{\alpha\beta}{\alpha+\beta})}.$$

$\square$

*Proof of Theorems 3 and 4.* Set $\mathcal{L}_q^{\text{vr}}(\varrho; f) := \mathbb{E}_{\mathbb{P}_{\text{true}}}[f] + \varrho\||\partial f|\|_q$. Observe the following decomposition

$$\begin{aligned}
&|\mathcal{L}_{n,p}^{\text{wo}}(\varrho; f) - \mathcal{L}_p^{\text{adv}}(\varrho; f)| \\
&\le |\mathcal{L}_{n,p}^{\text{wo}}(\varrho; f) - \mathcal{L}_{n,q}^{\text{vr}}(\varrho; f)| + |\mathcal{L}_{n,q}^{\text{vr}}(\varrho; f) - \mathcal{L}_q^{\text{vr}}(\varrho; f)| + |\mathcal{L}_q^{\text{vr}}(\varrho; f) - \mathcal{L}_p^{\text{adv}}(\varrho; f)|.
\end{aligned} \tag{11}$$

Below we bound the three terms on the right-hand side separately. For the last term, by [2, Remark 9], we have $|\mathcal{L}_p^{\text{wo}}(\varrho; f) - \mathcal{L}_q^{\text{vr}}(\varrho; f)| \le C\varrho^2$ for some $C \ge 0$.

We consider $p > 1$ first. For the first term, by Proposition 1, we have

$$|\mathcal{L}_{n,p}^{\text{wo}}(\varrho; f) - \mathcal{L}_{n,q}^{\text{vr}}(\varrho; f)| \le C_0\varrho^{(\alpha+1)\wedge p} + 2Le_n\big((pL/\tilde{\eta})^{q-1}\varrho\big).$$

For the second term, it follows from Lemma 3 and Lemma 5 that with probability at least $1 - 2e^{-t}$,

$$\begin{aligned}
|\mathcal{L}_{n,q}^{\text{vr}}(\varrho; f) - \mathcal{L}_q^{\text{vr}}(\varrho; f)| &\le |\mathbb{E}_{\mathbb{P}_{\text{true}}}[f] - \mathbb{E}_{\mathbb{P}_n}[f]| + \varrho|\||\partial f|\|_{q,\mathbb{P}_n} - \||\partial f|\|_{q,\mathbb{P}_{\text{true}}}| \\
&\le 2\mathfrak{R}_n(\mathscr{F}) + M\sqrt{\frac{t}{2n}} + \frac{\varrho}{q}(\eta \wedge \tilde{\eta})^{-\frac{q}{p}}\Big(2\mathfrak{R}_n(\partial\mathscr{F}_q) + L^q\sqrt{\frac{t}{2n}}\Big).
\end{aligned}$$

Therefore, we obtain the result.

Next, we consider $p = 1$. By Proposition 3,

$$|\mathcal{L}_{n,1}^{\text{wo}}(\varrho; f) - \mathcal{L}_{n,\infty}^{\text{vr}}(\varrho; f)| \le \epsilon_n.$$

Moreover, with probability at least $1 - e^{-t}$,

$$\begin{aligned}
|\mathcal{L}_{n,\infty}^{\text{vr}}(\varrho; f) - \mathcal{L}_\infty^{\text{vr}}(\varrho; f)| &\le \sup_{f \in \mathscr{F}}\{|\mathbb{E}_{\mathbb{P}_{\text{true}}}[f] - \mathbb{E}_{\mathbb{P}_n}[f]| + \varrho|\|G(f)\|_{\infty,\mathbb{P}_{\text{true}}} - \|G(f)\|_{\infty,\mathbb{P}_n}|\} \\
&\le 2\mathfrak{R}_n(\mathscr{F}) + M\sqrt{\frac{t}{2n}} + \Delta_n\varrho \\
&\le 2\mathfrak{R}_n(\mathscr{F}) + M\sqrt{\frac{t}{2n}} + \epsilon_n.
\end{aligned}$$

Finally, the third term is computed in Lemma 8. Thereby we complete the proof. $\square$