# OpenReview forum: "Generalization Bounds for (Wasserstein) Robust Optimization"
_NeurIPS.cc/2021/Conference — NeurIPS 2021 Poster_

### Official Review · Reviewer_HGjU · 2021-07-10

**Rating:** 7
**Confidence:** 3

**Summary:**

This paper studies the finite-sample generalization bounds for robust optimization and Wasserstein distributionally robust optimization. The results are more general than existing literature as they extend to piecewise Holder smooth loss functions. Besides the major contribution, the authors also provide application of the theorems in several examples, e.g. expected shortfall in Section 6.4 and adversarial training in Section 6.5. These examples show the usefulness of the main theorems.

**Limitations And Societal Impact:**

No social impact.

**Main Review:**

Originality: this paper has a clear literature review and compares the main results with the existing literature. The results in this paper are solid and novel.

Significance: this paper provides a lot of examples in various scenarios. These examples illustrate the significance of this paper to some extent. In addition, based on my knowledge, there is no existing literature that clearly studies the connection between distributional robustness and adversarial training. Improving the bounds is a contribution to this area, and utilizing it to other areas also brings more potentials.

Clarity and quality: this paper is well organized with clear illustrations. There is no experimental results, but numerical results are not essential for this paper.

**Time Spent Reviewing:**

5

---

> ### Author Response · Authors · 2021-08-10
> **Response to Reviewer HGjU**
>
> Thank you very much for taking the time to review our paper and for recognizing our contributions!

---

### Official Review · Reviewer_A21Q · 2021-07-12

**Rating:** 4
**Confidence:** 4

**Summary:**

The paper studies the generalization performance of minimizing Distributionally robust optimization (DRO) losses. In particular, it uses existing results that connect DRO to penalty (or regularization) based objectives and employs learning theoretic results to achieve concentration inequalities which ultimately manifest as a DRO generalization bound.

**Limitations And Societal Impact:**

Yes they have.

**Main Review:**

This submisison studies the generalization properties of DRO with quite interesting results by essentially merging different paradigms of proof techniques which appear nice conceptually. I believe the problem laid out is important however it is not clear to me how the offered bounds precisely resolves and adds anything actionable to the area of using DRO. Furthermore, this bleeds into my main concern which is that the results are highly inaccessible in their current form and difficult to understand their ramifications. I would prefer if the authors can clearly specify some benefits of their methods or existing work their methods provide validation for (specific methods as opposed to simply stating stochastic settings).

The correctness of the paper is of high standard as I have not found any mistake in the proofs after checking them and I find that all notation is self-contained which the authors should be applauded for given the dense machinery they have employed.

With regards to related work, they have focused on discussing work that relates to their proof techniques while briefly touching upon the adversarial training set-up. It seems they are missing quite a key reference since they have an entire paragraph of their related work discussing previous work on the link between robustness (DRO) and regularization yet a quick search with the words "distributional robustness and regularization" yields a NeurIPS2020 paper [1] which they don't seem to discuss. Other papers along this line can also be found for other cases which may be relevant such as https://www.jmlr.org/papers/volume10/xu09b/xu09b.pdf. Can the authors comment as to how the results in these papers relate to this or why it is irrelevant?

The applications they have are quite underwhelming since their motivation for a theoretical study is to better understand adversarial training (which often occurs in deep learning) yet their applications do not include a specific study on this - I would be interested to know any comments on this.

My other main concern is that the bounds contain several Rademacher complexities, some of which are highly inaccessible or difficult to grasp - my understanding is that Rademacher complexities are typically large and only for specific settings can be explicitly computed. For the case of neural networks, even a 2-layer neural network, do you have any comments on how the generalization bounds would look like? I'm especially interested to see the shape of Theorem 7.

Overall, while studying the generalization properties is important, I think the core theorems of the paper are quite simple combinations of existing DRO results and concentration inequalities which at the moment are unclear of their practical use for the motivated setting of adversarial robustness in deep learning. The specific analysis contains some more technical results however they are not motivated for the same problem and therefore fall short of a cohesive story. I will currently rate this as a reject however I am interested to see if the authors have any derivation for the deep learning setting or perhaps something more actionable for specific methods as opposed to explaining 'superior performance' of an entire paradigm they refer to as 'robust learning' - the ambiguity here is dangerous since there are many variants of robust learning which clearly have differing performances, not many of which the results at this stage comment on.

[1] Husain, H. (2020). Distributional Robustness with IPMs and links to Regularization and GANs arXiv preprint arXiv:2006.04349

**Time Spent Reviewing:**

10

---

> ### Author Response · Authors · 2021-08-10
> **Response to Reviewer A21Q**
>
> Thank you for raising up several important concerns that greatly help to clarify our contribution to the community. Indeed, there are various types of works investigating the generalization properties under the umbrella of robust learning and are easily confused with our setup. Below we will elaborate more on the significance, non-triviality, and implications of our results, and provide new results on the generalization bound for adversarial learning with neural networks.
>
> **1. On the contribution of the offered bounds added to DRO and ramification of our results.**
>
> Thank you for this important question, and we would like to take this opportunity to restate our technical contributions and their implications.
>
> (1) Section 3 establishes the relationship between robust optimization (RO), Wasserstein robust optimization (WO), and variation regularization (VR). They are preparatory results in order to derive our main results in Sections 4 and 5.
>
> - Theorem 1 shows that (WO) and (VR) are close with a gap controlled by two high-order terms: one depends on the Holder smoothness parameter $\alpha$; the other depends on the distribution of data samples around non-differentiable regions, which are shown to be small for losses such as $|\cdot|^p$ in Section 6.1.
> The take-away message here is that the smoother the loss function, the smaller the gap is; intuitively, the worst-case distribution in (WO) perturbs the data approximately aligned with gradient ascent direction, and gradient norm in (VR) provides a tighter first-order approximation for smoother losses.
>
> - Theorem 2 shows that (WO) and (RO) are close as the radius shrinks. Intuitively, both these two robust paradigms perturb data points aligned with the gradient ascent direction. The difference is that (WO) allows to perturb a tiny fraction of probability arbitrarily far away, but (RO) prohibits probability splitting, which results in a gap.
> To the best of our knowledge, the first result of this kind was established in Gao and Kleywegt (2016) for a single loss function but the bound therein can be undesirable for a family of losses.
> Here we derive a uniform gap between the two robust paradigms, using non-trivial careful analysis on the probabilistic nature of the worst-case distribution.
>
>
> - Theorem 3 is a counterpart result of Theorem 2 for 1-Wasserstein distance. It refines and improves the sandwich result in Gao et al. (2017) using careful probabilistic analysis on the worst-case distribution.
>
>
> (2) Section 4 presents our first set of main results that establish generalization bounds for (WO) and (RO) when data are generated stochastically, featured by that the radius shrinks to zero as more samples are available.
>
> - Both Theorem 4 and Theorem 5 shows that by choosing a reasonably small radius without suffering from the curse of dimensionality (as opposed to most Wasserstein DRO literature), the robust solution resulting from solving (WO) (and (RO)) has a good generalization error bound, expressed as the sum of the empirical loss and the empirical variation of the loss multiplied by the radius, plus a high-order term.
> The high-order term depends mainly on the smoothness of the loss function, the (non-)concentration of probability distributions on non-differentiable points, and the complexity of the gradient norm function class (when the Wasserstein order $p>1$).
>
> - Various examples are provided in Section 6 to demonstrate different perspectives of our results: non-smooth losses (Section 6.1 and 6.4), non-i.i.d. (Section 6.3), and improved robustness-based generalization bound compared to the literature (Sections 6.2-6.4).
>
> - We would like to emphasize that this setting is drastically *different* from the adversarial setting where many recent studies on the generalization properties of robust optimization have been focused on.
> In fact, choosing a good radius scaling scheme with nice finite-sample guarantees is at the core of DRO and remains open for quite some time until the recent work of Gao (2020). We generalize this result to non-i.i.d., non-smooth settings as well as for (RO); see Line 55-65 for more detailed comparison and Line 66-78 for other related works.
>
> (3) Section 5 presents our second set of main results that establish generalization bounds for (WO) when data are generated adversarially, featured by that the radius is a fixed, tiny number.
>
> - While there are many recent works studying this problem especially $\infty$-Wasserstein DRO (or equivalently (RO)), we do not study the adversarial Rademacher complexity as several existing works did, but adopt a different treatment on the basis of our preparatory results in Section 3.
>
> - Different from those existing works, our bound involves a term reflecting the complexity of the gradient norm function class, which appears to be new in the literature and comes from our tools on regularization.
> This makes intuitive sense as in the adversarial setting, the worst-case loss is approximately equal to the nominal loss plus a gradient penalty, thereby the complexity of the gradient penalty plays a role in generalization. In the subsequent response below, we will show that our new results indeed provide a dimension-independent generalization bound for adversarial learning with a neural network, which significantly improves the existing results.
>
> **2. On related works on distributional robustness and regularization.**
>
> The reason why we include the discussion on distributional robustness and regularization is that part of our analysis relies on it. Nonetheless, by no means do we claim this is our main originality---our main contribution is on deriving the generalization properties of robust optimization and our new regularization results are used for establishing them. We completely agree that there are tons of research studying the connection between distributional robustness and regularization. In the introduction, we mainly discuss papers that focus on robust optimization and Wasserstein robust optimization, and we apologize for omitting any important ones.
> Particularly to your point, in Line 88 we have cited Xu et al. (2009) referenced by your webpage link, which is indeed one of the pioneer works on this topic. We will cite Husain H. (2020) in our revision as it provides a unified analysis of the connection between various distributional robustness and regularization.

---

> > ### Author Response · Authors · 2021-08-10
> > **On the exemplification of Theorem 7 for neural networks**
> >
> > Below we provide an example of Theorem 7 in Section 6.5 where $\\mathscr F$ is a neural network, using the same setup as in Section 5 of Awasthi et al. (2020).
> > Let $\\sigma$ be an activation function, which is assumed to have Lipschitz norm $L_\\sigma$ and has Lipschitz gradient with norm $h_\\sigma$. Consider the function class $\\mathscr F$ on the domain $\\{ x \\in \\mathbb R^d:\\ \\lVert x \\rVert_2 \\leq B_x\\}$ defined as
> > \\[
> > \\begin{aligned}
> > \\mathscr F &= \\{ x \\mapsto u^\\top \\sigma(Wx): (u,W)\\in\\Theta\\},
> > \\end{aligned}
> > \\]
> > here
> > $$
> > \\Theta :=\\{(u,W): u\\in \\mathbb{R}^m, W \\in \\mathbb{R}^{m \\times d},  \\lVert u \\rVert_1 \\leq B_u, \\lVert W \\rVert_{2, \\infty} \\leq B_W\\},
> > $$
> > and $\\lVert W \\rVert_{2, \\infty} = \\max_{k = 1}^m \\lVert W_k\\rVert_{2}$, where $W_k$ is the $k$-th row of $W$.
> >
> > In light of the setting of Section 6.5, we are interested in bounding the term $\\mathfrak R_n(\\partial \\mathscr F_1^\\phi)$, where
> > \\[
> > \\begin{aligned}
> > \\partial \\mathscr F_1^\\phi &= \\left\\{(x, y) \\mapsto \\vert \\phi'(yu^\\top \\sigma(Wx))\\vert \\Vert \\sum_{k =1}^m u_k  \\sigma'(W_k x) W_k \\Vert_2: (u,W)\\in\\Theta \\right\\}.
> > \\end{aligned}
> > \\]
> >
> >
> > To begin with, we write the gradient norm function as
> > \\[
> > \\vert \\phi'(yu^\\top \\sigma(Wx))\\vert \\Vert \\sum_{k =1}^m u_k \\sigma'(W_k x) W_k \\Vert_2 = g \\Big( g_1(x, y;u,W), g_2(x, y;u,W) \\Big),
> > \\]
> > where $g$ is the scalar product function $g(c_1, c_2) := c_1 c_2$, and
> > \\[
> >   g_1(x, y;u,W) := \\vert \\phi'(yu^\\top \\sigma(Wx))\\vert,\\quad
> >   g_2(x, y;u,W) := \\Vert \\sum_{k =1}^m u_k \\sigma'(W_k x) W_k  \\Vert_2.
> > \\]
> > Observe that the function $g$ has Lipschitz constant $\\sqrt{2}(L_\\phi + L_\\sigma B_W B_u)$ on the range of $g_1,g_2$.
> > Hence, using the vector-contraction inequality for Rademacher complexity [1], we have
> > \\[
> >   \\mathfrak R_n (\\partial \\mathscr F_1^\\phi) \\le 2(L_\\phi + L_\\sigma B_W B_u) \\Big(\\mathfrak R_n(\\{g_1(\\cdot;u,W):(u,W)\\in\\Theta\\}) + \\mathfrak R_n(\\{g_2(\\cdot;u,W):(u,W)\\in\\Theta\\})\\Big).
> > \\]
> > The first Rademacher complexity can be upper bounded by
> > \\[
> > \\mathfrak R_n \\Big(\\{g_1(\\cdot;u,W):(u,W)\\in\\Theta\\})\\Big) \\le h_\\phi \\mathfrak R_n(\\mathscr F) \\le \\frac{h_\\phi L_\\sigma B_W B_x}{\\sqrt{n}}.
> > \\]
> > using standard result for the Rademacher complexity of neural networks. Therefore, in the sequel we focus on bounding the second Rademacher complexity. Let
> > \\[
> >   R := \\mathbb E_\\epsilon \\Big[  \\sup_{\\lVert W \\rVert_{2, \\infty} \\leq B_W} \\sup_{  \\lVert u \\rVert_1 \\leq B_u} \\sum_{i = 1}^n \\epsilon_i  \\lVert \\sum_{k =1}^m u_k  \\sigma'(W_k x_i) W_k \\rVert_2 \\Big],
> > \\]
> > where $\\epsilon_i$'s are i.i.d.~Rademacher random variables.
> > Observe that
> > \\[
> >   \\sup_{  \\lVert u \\rVert_1 \\leq B_u} \\sum_{i = 1}^n \\epsilon_i  \\lVert \\sum_{k =1}^m u_k  \\sigma'(W_k x_i) W_k \\rVert_2
> > \\]
> > maximizes a convex function of $u$ over a convex region $\\{u\\in\\mathbb R^m: \\lVert u \\rVert_1 \\leq B_u\\}$, hence its supremum is attained at the extreme points of the region, thus
> > \\[
> >   \\sup_{  \\lVert u \\rVert_1 \\leq B_u} \\sum_{i = 1}^n \\epsilon_i  \\lVert \\sum_{k =1}^m u_k  \\sigma'(W_k x_i) W_k \\rVert_2 = \\max_{k = 1, \\ldots m} B_u \\sum_{i = 1}^n \\epsilon_i  \\lVert   \\sigma'(W_k x_i) W_k \\rVert_2.
> > \\]
> > It follows that
> > \\[
> >   R \\le B_u \\mathbb E_\\epsilon \\Big[ \\sup_ {\\lVert A \\rVert_{2, \\infty} \\leq B_W} \\max_{k = 1, \\ldots m} \\sum_{i = 1}^n \\epsilon_i  \\lVert   \\sigma'(W_k x_i) W_k \\rVert_2 \\Big].
> > \\]
> > Observe that the above maximization attains its optimal value when
> > \\[
> >   A_k \\in \\arg\\max_{\\lVert w \\rVert_2 \\leq B_W}  \\sum_{i = 1}^n \\epsilon_i  \\lVert   \\sigma'(w^\\top x_i) w \\rVert_2,\\quad 1\\le k\\le m,
> > \\]
> > where $w\\in\\mathbb R^d$.
> > It follows that
> > \\[
> > R \\le B_u \\mathbb E_\\epsilon \\Big[\\sup_ {\\lVert w \\rVert_2 \\leq B_W}  \\sum_{i = 1}^n \\epsilon_i  \\lVert   \\sigma'(w^\\top x_i) w \\rVert_2 \\Big].
> > \\]
> > Notice that $\\lVert  \\sigma'(w^\\top x) w \\rVert_2 = g(\\lVert w \\rVert_2, \\vert \\sigma'(w^\\top x) \\vert)$, where the product function $g$ has Lipschitz constant $\\sqrt{2}(B_W + L_\\sigma)$ on the range determined by $w$ and $\\sigma'$.
> > Using again the vector-contraction inequality for Rademacher complexity [1], we have
> > \\[
> > \\begin{aligned}
> > &\\quad \\mathfrak R_n(\\{x \\mapsto \\lVert   \\sigma'(w^\\top x) w \\rVert_2: \\lVert w \\rVert_2 \\leq B_W\\})\\\\
> > &\\leq 2(B_W + L_\\sigma) \\Big(\\mathfrak R_n(\\{x \\mapsto \\lVert  w  \\rVert_2: \\lVert w \\rVert_2 \\leq B_W\\}) + \\mathfrak R_n(\\{x \\mapsto \\vert \\sigma'(w^\\top x) \\vert: \\lVert w \\rVert_2 \\leq B_W\\}) \\Big)\\\\
> > &\\leq 2(B_W + L_\\sigma) \\big( \\frac{B_W}{\\sqrt{n}} + \\frac{h_{\\sigma} B_W B_x}{\\sqrt{n}}\\big),
> > \\end{aligned}
> > \\]
> > where the last equality follows from standard approach.
> > Therefore, we finally have
> > \\[
> > \\begin{aligned}
> > \\mathfrak R_n(\\partial \\mathscr F_1^\\phi) \\leq 2(L_\\phi + L_\\sigma B_W B_u) \\Big(h_{\\phi}L_\\sigma B_W B_x + 2(B_W + L_\\sigma) (B_W + h_{\\sigma} B_W B_x) \\Big) \\frac{1}{\\sqrt{n}}.
> > \\end{aligned}
> > \\]
> > Using the displayed equation below line 343, the above result implies that the generalization bound of adversarial learning using neural network $\\mathscr F$ is dimension-independent, i.e, independent of the dimension of input and the width of the neural net, and thereby significantly improves the $O(\\sqrt{md})$ bound in Theorem 7 of Awasthi et al. (2020) thus resolve an open question raised in their Section 6. The only catch here is that our bound has an additional $O(\\rho^2)$ term, which is typically very small in adversarial learning problems.
> > Of course, this is not the end of the story and we plan to study the generalization bound of adversarial learning using multi-layer neural network in our follow-up work.
> > We believe our framework can provide promising results in this direction.
> >
> > Finally, we would like to remark that adversarial learning is only one of the main motivation of our study, namely, the results in Sections 5 and 6.5. Typically in these problems, the Wasserstein radius is fixed. On the other hand, our results in Sections and 6.1-6.4 does not consider adversarial settings, and thus are motivated by other applications of Wasserstein robust learning.
> >
> > **Reference**
> >
> > [1] Andreas Maurer. A vector-contraction inequality for rademacher complexities. In International Conference on Algorithmic Learning Theory, pages 3–17. Springer, 2016.

---

> ### Author Response · Authors · 2021-08-24
> **Thanks for your time and I hope our reponse helps the re-assessment!**
>
> Dear Reviewer A21Q,
>
> We sincerely hope our reponse on your comments can help answer your quesitions. If you have any further comments, or anything that needs clarifications, please don't hesistate to let us know, and we are happy to write follow-up reponses.
>
> Thank you very much for your time!

---

### Official Review · Reviewer_XTX2 · 2021-07-12

**Rating:** 6
**Confidence:** 4

**Summary:**

This paper aims to theoretically analyze (and compare) a number of different robust learning paradigms. In particular, it has been shown that three different notions of robustness, namely 1) robust optimization 2) Wasserstein distributionally robust optimization, and 3) variation regularization, would result in close loss functions (based on some non-asymptotic bounds) whenever the strength of attacks becomes infinitesimally small. While connections between such methods have been conjectured (or even analyzed) in some recent works, I still found the results in this paper interesting.

Also, author(s) have tried to achieve non-asymptotic generalization bounds for cases where data samples are weakly-dependent (instead of i.i.d. generated) and/or the space of loss functions are only piece-wise smooth. I have not checked the proofs for this part, and results have not been clearly discussed nor compared with existing ones.

Overall, a well-motivated paper which tries to add a number of nice contributions to the area. However, paper suffers from lack of clarifications and intuitive discussions in order to shed light on the technical significance of its many theorems. Also, some notations in parts of the paper (including theorem 2) are not properly framed. Another problem with this work is that author(s) might have missed many existing works that have already tried to analyze the same (or at least slightly different) problem sets. As a result, paper is not well-positioned w.r.t. the literature and cannot be easily compared with other researches.

I cannot recommend the paper for acceptance in its current shape, thus my vote at this stage is weak reject.


**Limitations And Societal Impact:**

No problem here.

**Main Review:**

Strengths:

Paper is well-motivated. The fact that data samples can be generated in a weakly-dependent manner instead of the common i.i.d. assumption is nice. Also, considering piece-wise smooth functions instead of globally smooth ones is interesting for many applications. Moreover, author(s) have put so much effort to write the paper in a mathematically comprehensive way. This has (unfortunately) reduced the readability of the paper, but has made it technically rigor (except for a few parts, described below). By the way, I have not checked all the proofs.

----------------------------------------------------------------------------------------------------------------

Weaknesses: There are several issues with the manuscript

1) Paper is well-organized, but when someone goes into details it is not written in an informative way. In many cases, several layers of nested definitions have been made with few to no intuition for the reader. Understanding the statement of main theorems requires one to go through all these cryptic definitions with so many parameters and variables. This problem has reduced the clarity of results and has made the paper hard to read. A total of 8 assumptions have been made throughout the main manuscript (haven't completely checked the supplements), almost all mathematically encrypted. And authors have not provided enough discussion on why making such assumptions could be plausible in real-world. One reason for all the above-mentioned issues might be due to the fact that author(s) have tried to fit so much content in a page-limited manuscript. Therefore, I kindly suggest to the author(s) to consider sending the paper to a journal instead of a conference.


2) Paper is not well-positioned w.r.t. many recent works which have also addressed the generalization aspects of adversarial robustness in a distributionally robust setting. For example, Sinha et al. (2017), Carmon et al. (2019), Najafi et al. (2019), and several other works have recently proposed generalization guarantees for DRL (a.k.a. DRO) where generalization gap is $O\left(n^{-1/2}\right)$, similar to Theorem 6 of the current work. What has been improved in this work compared to existing results? This question is either left unanswered in this work, or the significance of the current results is buried under tones of cryptic and non-intuitive definitions and statements. In the latter case, author(s) should point out to the potential improvements and/or fundamental differences.


3) There are some technical ambiguities that need to be addressed:
-Theorem 1 is interesting. However, author(s) have not compared it well enough w.r.t. to existing works in the literature. The fact that distributionally robust learning has connections to some types of smoothness regularization over the hypothesis space is already known to the community (For example, check the keyword "distributional smoothing"). What is the main advancement in this particular theorem? For example, how is it different from Gao et al. (2017), also cited by the author(s), since I noted a rather similar result in Lemma 1 of that work.


-Paper seems to be rushed in some parts. For example, Theorem 2 is supposed to be one of the main contributions, still its statement is not correctly framed: What is parameter $t$? Does having $\varrho_n\rightarrow 0$ when $n\rightarrow\infty$ suffice? does the rate of convergence toward zero matter? I assume they all affect the integer $N$, however, such issues need to be clarified. Otherwise, author(s) cannot claim the bound in this theorem is "non-asymptotic" since $N$ can become uncontrollably large.

---------------------------------------------------------------------------
Minor comments:

-(Line 67): the word "generalization" is repeated.
-(Line 118): This line doesn't make sense, unless one assumes $\inf_{y\in\mathcal{D}_f}\left\Vert x-y\right\Vert$ instead of what has been written. Also, a full stop is missing.

**Time Spent Reviewing:**

5 hours

---

> ### Author Response · Authors · 2021-08-10
> **Response to Reviewer XTX2 -- first part**
>
> Thank you very much for your constructive comments.
> We are committed to improve the clarity in the revised version, and below we would like to provide more intuitive discussions and comparisons.
>
> **1. On the presentation of our results.**
>
> Thank you pointing out this issue. Indeed, as you said, we sacrifice some readability for mathematical rigor.
> Perhaps the reasons why one may find difficulty in understanding our results are that (1) we squeeze our comparisons with existing literature in the introduction without restating them in the main text, and (2) that we postpone our illustrative examples after all theories are presented and the explanations are scattered in various places in the paper that are not easy to recognize in a short time.
> We appreciate your understanding and below we would like to explain every result in our paper in more detail.
>
> (1) Section 3 establishes the relationship between robust optimization (RO), Wasserstein robust optimization (WO), and variation regularization (VR).
>
> - Theorem 1 shows that (WO) and (VR) are close with a gap controlled by two high-order terms: one depends on the Holder smoothness parameter $\alpha$; the other depends on the distribution of data samples around the non-differentiable regions, which are shown to be small for losses such as $|\cdot|^p$ in Section 6.1.
> The take-away message here is that the smoother the loss function, the smaller the gap is; intuitively, the worst-case distribution in (WO) perturbs the data approximately aligned with gradient ascent direction, and gradient norm in (VR) provides a tighter first-order approximation for smoother losses.
> The assumptions are standard and simplify those made in Gao et al. (2017); see bullet point 3. (1) below for a more detailed comparison.
>
> - Theorem 2 shows that (WO) and (RO) are close as the radius shrinks. Intuitively, both these two robust paradigms perturb data points aligned with gradient ascent direction. The difference is that (WO) allows to perturb a tiny fraction of probability arbitrarily far away, but (RO) prohibits probability splitting, which results in a gap.
> To the best of our knowledge, the first result of this kind was established in Gao and Kleywegt (2016) for a single loss function but the bound therein can be undesirable for a family of losses.
> Here we derive a uniform gap between the two robust paradigms, using non-trivial careful analysis on the probabilistic nature of the worst-case distribution.
> We presented a general result that allows an arbitrary rate of shrinking, thereby the big $N$ in the theorem depends on the probability confidence level $t$ and other parameters implicitly, as well as additional assumptions regarding the (non-)concentration of true probability distribution around non-differentiable points and the complexity of the gradient norm function class; see bullet point 3.(2) below for a refined result on the rate of convergence.
>
>
> - Theorem 3 is a counterpart result of Theorem 2 for 1-Wasserstein distance. It refines and improves the sandwich result in Gao et al. (2017) using careful probabilistic analysis on the worst-case distribution. Explanations on the new Assumption 5 is provided in line 181-187.
>
>
> (2) Section 4 presents our first set of main results that establish generalization bounds for (WO) and (RO) when data are generated stochastically, featured by that the radius shrinks to zero as more samples are available.
>
> - Both Theorem 4 and Theorem 5 shows that by choosing a reasonably small radius without suffering from the curse of dimensionality (as opposed to most Wasserstein DRO literature), the robust solution resulting from solving (WO) (and (RO)) has a good generalization error bound, expressed as the sum of the empirical loss and the empirical variation of the loss multiplied by the radius, plus a high-order term.
> The high-order term depends mainly on the smoothness of the loss function, the (non-)concentration of probability distributions on non-differentiable points, and the complexity of the gradient norm function class (when the Wasserstein order $p>1$). The assumptions generalizes those in Gao (2020) to non-iid settings.
>
> - Various examples are provided in Section 6 to demonstrate the different perspectives of our results: non-smooth losses (Section 6.1 and 6.4), non-i.i.d. (Section 6.3), and improved robustness-based generalization bound compared to the literature (Sections 6.2-6.4).
>
> - We would like to emphasize that this setting is drastically *different* from the adversarial setting where many recent studies on the generalization properties of robust optimization have been focused on.
> In fact, choosing a good radius scaling scheme with nice finite-sample guarantees is at the core of DRO and remains open for quite some time until the recent work of Gao (2020). We generalize this result to non-i.i.d., non-smooth settings as well as for (RO); see line 55-65 for more detailed comparison and line 66-78 for other related works.
>
> (3) Section 5 presents our second set of main results that establish generalization bounds for (WO) when data are generated adversarially, featured by that the radius is a fixed, tiny number.
>
> - While there are many recent works studying this problem especially $\infty$-Wasserstein DRO (or equivalently (RO)), we do not study the adversarial Rademacher complexity as several existing works did, but adopt a different treatment on the basis of our preparatory results in Section 3. See more history on this topic and detailed comparison with related works in bullet point 2 below.
>
> - Different from those existing works, our bound involves a term reflecting the complexity of the gradient norm function class, which appears to be new in the literature and comes from our tools on regularization.
> This makes intuitive sense as in the adversarial setting, the worst-case loss is approximately equal to the nominal loss plus a gradient penalty, thereby the complexity of the gradient penalty plays a role in generalization. In our response to Reviewer  A21Q, we show that our new results indeed provide a dimension-independent generalization bound for adversarial learning with a two-layer neural network, which significantly improves the existing results.

---

> > ### Author Response · Authors · 2021-08-10
> > **Response to Reviewer XTX2 -- second part**
> >
> > **2. On the positioning of the results relative to other works on adversarial robustness.**
> >
> > Thank you for bring these papers to our attention. Indeed, as you pointed out, there are many recent works aiming at addressing the generalization aspects of adversarial robustness in the distributionally robust setting, such as Lee and Raginsky (2018) for $p\in[1,\infty)$; Sinha et al. (2017) and Najafi et al. (2019) for $p=1$; and Schmidt et al. (2018), Tsipras et al. (2018), Bubeck et al. (2019), Zhang et al. (2019), Khim and Loh (2018), Yin et al. (2019), Awasthi et al. (2020) for $p=\infty$.
> > We briefly discuss some of them in line 79-93 in the introduction, but we would like to take this opportunity to compare our results with them in more details, especially for the references that you have pointed out.
> >
> > Lee and Raginsky (2018) is the first work to consider a similar settings as ours with $p\in[1,\infty)$. The main difference is that we consider different magnitudes of data perturbation. We mainly focus on adversarial robustness where the radius is often a tiny number, and our results in Section 5 suggest that the generalization gap is small when the radius is small.
> > While their motivation was from domain adaptation, hence their bounds is mostly useful when the Wasserstein radius is not extremely small, and their results suggest that the generalization gap becomes smaller as the radius grows to infinity (see their Remark 4).
> >
> > Sinha et al. (2017) and Najafi et al. (2019) focuses on $p=1$.
> > The apparent difference between Sinha et al. (2017) and ours is that their bounds in Theorem 3 and Corollary 2 are based on the covering number while our bound is expressed in terms of Rademacher complexity, which is often tighter.
> > Moreover, the term $\epsilon_n(t)$ in their equation (12) depends on the value of the variable $\gamma$, corresponding to the Lagrangian multiplier of the Wasserstein distance constraints in the dual formulation of Wasserstein DRO. An upper bound derived in their Corollary 2 does not involve the dependence on the Wasserstein radius $\rho$; so does the bound in Theorem 3 of Najafi et al. (2019).
> > In contrast, our Theorem 6 indicates that the remainder actually scales linear in $\rho$. In practice, $\rho$ is often a very small number, therefore our bound provides a tighter constant and demonstrates its scaling in the Wasserstein radius.
> >
> > Khim and Loh (2018), Yin et al. (2019), Awasthi et al. (2020) focuses on deriving the generalization bounds for specific loss function classes such as linear hypotheses and neural networks when $p=\infty$. In Section 6.5, we showed that our bound agrees with the bound in Awasthi et al. (2020) for linear class. Our results apply to generic function classes, and most importantly, the bound in Theorem 7 suggests the the generalization gap is controlled by not only the complexity of the loss function class, but also the complexity of the gradient norm functions of the loss functions. The intuition is that the gradient norm controls the robustness of the model subject to adversarial perturbations; see line 271-277 for more details. As far as we know, the bound in such form is new in the literature.
> >
> > There are also some earlier works focusing on investigating the fundamental limit of adversarial learning and its difference from empirical risk minimization, such as Schmidt et al. (2018), Bubeck et al. (2018), Tsipras et al. (2018), and Carmon et al. (2019). Their consider somewhat different settings than ours, which usually involves a stylized data generating model that facilitate theoretical findings.
> >
> > Finally, we remark that all these works on adversarial robustness consider a fixed radius, while we also consider the regime where the radius shrinks when more data is available, which is referred to as the stochastic setting in our paper.
> >
> > **3. On some techincal ambiguities.**
> >
> > Thank you for rising up these issues that help to clarify our results.
> >
> > (1) The main difference between Theorem 1 and Gao et al. (2017) is that we impose different assumptions on the loss functions as hinted in line 157-158. More specifically, in Theorem 1, by focusing on Lipschitz loss functions, we are able to obtain a cleaner result comparing to Theorem 1 in Gao et al. (2017), without extra assumptions on the density of the distribution or the growth of the gradient (Assumptions 3 and 4 in Gao et al. (2017)).
> > Besides, Lemma 1 in Gao et al. (2017) considers only smooth losses.
> > We completely agree that there are existing works on distributional smoothing and regularization, but we would also like to remark that Theorem 1 is just a preparatory result for our main development of generalization bounds in Sections 4 and 5.
> >
> > (2) This is an excellent point! $t$ is indeed hidden in $N$, which does depend on the rate of convergence of the relevant terms. We gave a general result in the current paper, but once we have the rate of convergence on those relevant terms, we can explicitly compute $N_0$, such that $N > N_0$ would be sufficient. For example, suppose $\\mathfrak R_S(\\partial \\mathscr F_q) \\leq \\frac{C_1}{n^{s_1}}$, $\\frac{e_n((\\frac{2L}{\\eta})^{q - 1} \\varrho_n)}{(\\frac{2L}{\\eta})^{q - 1} \\varrho_n} \\leq \\frac{C_2}{n^{s_2}}$, and $\\varrho_n \\leq \\frac{C_3}{\\sqrt{n}}$ for some constants $s_1, s_2, C_1, C_2, C_3$, and all $n \\in \\mathbb{N}^+$.
> > Then $N_0$ can be chosen so that Line 563 in Appendix holds, and that $\\lVert |\\partial f| \\rVert_{q, \\mathbb{P}_n}\\ge \\eta /2$ with probability at least $1- e^{-t}$ via Lemma 1.
> >
> > More explicitly, one can choose
> > \\[
> > \\begin{aligned}
> > N_0 &= (\\frac{9C_12^{q - 1}}{q\\eta^{q - 1}})^{\\frac{1}{s_1}} + 2\\Big( \\frac{3L 2^{q-1}}{q\\eta^{q-1}} \\Big)^2t + \\frac{7Lt2^{q-1}}{q\\eta^{q-1}}\\\\
> > &+ \\Big( \\frac{40L (2L)^{q-1}C_2}{\\eta^{q}} \\Big)^{\\frac{1}{s_2}} +C_3^2 \\Big( \\frac{L^{q-1 -\\frac{1}{\\alpha}}h^{\\frac{1}{\\alpha}}10^{q-1}}{\\eta^{q-1}} \\Big)^2 + \\Big( \\frac{20LphC_3}{\\eta^{2}} \\Big)^2\\\\
> > &+ C_3^2\\Big( \\frac{2^{\\alpha + 1}10h}{\\eta} \\Big)^{\\frac{2}{\\alpha}} + \\Big( \\frac{10\\cdot 2^{p}}{\\eta} \\Big)^{2(q-1)}C_3^2,
> > \\end{aligned}
> > \\]
> > where $\\eta = \\inf_{f\\in\\mathscr F} \\lVert |\\partial f| \\rVert_{q, \\mathbb{P}_{\\mathsf{true}}} > 0$ as in Assumption 4; $L$ is the Lipchitz constant as in Assumption 1; $h, \\alpha$ comes from Assumption 2; and $t$ is the arbitrarily chosen positive number that reflects the confidence level in Theorem 2.
> >
> > As we see, $N_0$ is linear in $t$, so if we want a higher probability, we also need a larger $N_0$.

---

> ### Author Response · Authors · 2021-08-24
> **Thanks for your time and I hope our reponse helps the re-assessment!**
>
> Dear Reviewer XTX2,
>
> We sincerely hope our reponse on your comments can help answer your quesitions. If you have any further comments, or anything that needs clarifications, please don't hesistate to let us know, and we are happy to write follow-up reponses.
>
> Thank you very much for your time!

---

> > ### Comment · Reviewer_XTX2 · 2021-08-26
> > **I raise my vote to weak-accept**
> >
> > Dear authors,
> >
> > Thank you for your comprehensive answer. I can say that the majority of my concerns are alleviated after reading your response and also going through other reviewers' comments. In any case, I still have two concerns which prevent me from raising my score above a weak-accpet:
> >
> > First, comparison with distributionally robust settings, and in particular those highlighted in Sinha et al. (2017) and Najafi et al. (2019), are still vague. I disagree with parts of your response, since none of those works restrict $p$ to be $1$, and the majority of other smoothness constraints are solely used to mitigate the computational complexity of the problem, and not its sample complexity bounds. Also, Najafi et al. (2019) uses Rademacher complexity like your work, and not covering numbers. Both works, either implicitly (Sinha et al.) or explicitly (Najafi et al.) have shown that Rademacher complexity can be bounded regardless of the Wasserstein radius, which means any learnable model is also adversarially learnable. I agree with the authors that using the Wasserstein radius $\rho$ explicitly, instead of its dual counterpart $\gamma$, should be considered as an improvement, but it is still a marginal edge over those works.
> > In general, the main point of weakness is that reader might not be able to see how your paper has improved, or maybe even contrasted any of the results in this particular line of research. As I said earlier, it makes it hard to position the submission w.r.t. existing literature.
> >
> > The other issue is clarity. I appreciate the fact that authors also agreed about this problem. However, I still do not see how it can be resolved. In other words, paper does not convey a clear message and is loaded with tones of hard-to-read mathematical results.
> >
> > I raise my vote to weak accept, since I am more certain about the technical validity of results now.

---

> > > ### Author Response · Authors · 2021-08-27
> > > **Thanks so much for your feedback!**
> > >
> > > We apologize for the vague comparison with the two references. We took another look at the two references we agree mostly with what you said. We still believe our improvement on the linear scaling of the remainder scales in $\rho$ is crucial for small $\rho$, which enables nice generalization bounds as in our response to Reviewer A21Q.
> > >
> > > We will add those comparisons as well as intuitive explanations as in our previous reply to the revised paper, and we hope it helps increase the clarity of our results.
> > >
> > > Thanks again for taking the time to review our paper!

---

### Official Review · Reviewer_vWvJ · 2021-07-16

**Rating:** 9
**Confidence:** 5

**Summary:**

This paper derives finite-sample generalization bounds for robust optimization and Wasserstein distributionally robust optimization. The authors consider a broad class of piecewise Hölder smooth loss functions, under both stochastic setting—i.i.d. or weakly dependent data—and adversarial environment, assuming that the underlying data-generating distribution satisfies transportation-information inequalities. The proofs are built on a general connection between robustness and variation regularization (including Lipschitz and gradient regularization, among others) and new local Rademacher complexity results for variation regularization.

**Limitations And Societal Impact:**

Yes

**Main Review:**

1. Besides the  Wasserstein distributionally robust optimization (WO) problem，this paper considers the robust optimization (RO) problem and establishes new general bounds. Specifically, in the stochastic setting, under proper choice of the radius $\varrho_n$, this paper derive finite-sample guarantees for (RO) and (WO) that apply to non-i.i.d. data and non-smooth losses. For non-i.i.d. data, this paper exploits different notions of transport inequality and Rademacher complexity than the i.i.d. case. In the adversarial setting, generalization bounds for (Adv) with fixed small radius $\varrho_n$ are derived.

2. The analysis of this paper is based on the connection between robustness and regularization. The proposed results improve the sandwich bounds in Gao et al. (2017) for $p = 1$, and generalizes the results therein to piecewise Hölder smooth loss functions for $p\in (1,\infty].


**Time Spent Reviewing:**

12

---

> ### Author Response · Authors · 2021-08-10
> **Response to Reviewer vWvJ**
>
> Thank you for your time in reviewing our paper, and for your precise summary of our contribution comparing to existing literature. We appreciate your positive recognition of results.

---

### Decision · Program_Chairs · 2021-09-27

**Decision:**

Accept (Poster)

**Comment:**

Overall, the paper contains interesting technical contributions on generalization bounds for DRO problems. However, the technical exposition makes it difficult to access the results in the paper and has left much to be desired. In particular, the motivations and technical development need to be substantially improved.